## Registered report

psychology

dyslexia, averaging, ensemble coding, perception, internal noise, magnocellular

**Author for correspondence:**
Catherine Manning
e-mail: c.a.manning@reading.ac.uk

# Integration of visual motion and orientation signals in dyslexic children: an equivalent noise approach

Catherine Manning[1,2], Victoria Hulks[1], Marc S. Tibber[3] and Steven C. Dakin[4,5]

[1]Department of Experimental Psychology, University of Oxford, UK
[2]School of Psychology and Clinical Language Sciences, University of Reading, UK
[3]Department of Clinical, Educational and Health Psychology, UCL, UK
[4]School of Optometry and Vision Science, University of Auckland, New Zealand
[5]UCL Institute of Ophthalmology, University College London, UK

CM, 0000-0001-6862-2525; SCD, 0000-0002-3548-9104

Dyslexic individuals have been reported to have reduced global motion sensitivity, which could be attributed to various causes including atypical magnocellular or dorsal stream function, impaired spatial integration, increased internal noise and/or reduced external noise exclusion. Here, we applied an equivalent noise experimental paradigm alongside a traditional motion-coherence task to determine what limits global motion processing in dyslexia. We also presented static analogues of the motion tasks (orientation tasks) to investigate whether perceptual differences in dyslexia were restricted to motion processing. We compared the performance of 48 dyslexic and 48 typically developing children aged 8 to 14 years in these tasks and used equivalent noise modelling to estimate levels of internal noise (the precision associated with estimating each element's direction/orientation) and sampling (the effective number of samples integrated to judge the overall direction/orientation). While group differences were subtle, dyslexic children had significantly higher internal noise estimates for motion discrimination, and higher orientation-coherence thresholds, than typical children. Thus, while perceptual differences in dyslexia do not appear to be restricted to motion tasks, motion and orientation processing seem to be affected differently. The pattern of results also differs from that previously reported in autistic children, suggesting perceptual processing differences are condition-specific.

# 1. Introduction

Dyslexia is a neurodevelopmental condition characterized by difficulties in learning to read and spell [1,2]. While deficits in phonological processing appear to be the largest cognitive factor contributing to reading difficulties in dyslexia [3,4], a range of other cognitive deficits have been proposed, including visual processing differences (see [5], for review). While most research suggests that visual processing differences do not have a causal role in the development of dyslexia ([4], but see also [6]), it seems that there are nonetheless group differences between dyslexic individuals and normal readers that may reflect atypical brain development [7].

One aspect of visual processing that has received much attention in dyslexia is global motion processing. Typically, this is assessed by asking participants to detect or discriminate the direction of coherently moving 'signal' dots amidst a field of randomly moving 'noise' dots. On average, dyslexic individuals require more dots to be moving coherently in order to perceive the overall motion compared to individuals without dyslexia (e.g. [8–10]; see also [11], for meta-analysis). However, the reason for dyslexic individuals' difficulties with coherent motion processing remains uncertain, with at least four explanations in the literature.

First, elevated motion-coherence thresholds could result from atypical functioning in the magnocellular system [12–14] and/or dorsal stream [15], potentially explaining why performance in motion-processing tasks is disproportionately affected in dyslexia compared to tasks that rely less heavily on these systems, such as static global form processing tasks [8,16,17]. Second, elevated motion-coherence thresholds could arise from difficulties in filtering out the randomly moving noise dots (noise exclusion: [18,19]). Third, elevated motion-coherence thresholds could arise from higher levels of internal noise in dyslexic individuals [20–23], leading to less precise estimates of each dot's direction [24]. Finally, elevated motion-coherence thresholds could arise from poor integration of motion signals, which may be owing to spatial undersampling resulting from fewer motion detectors in dyslexic individuals than non-dyslexic individuals [9]. These explanations are not necessarily mutually exclusive, as reduced sampling could be specific to the magnocellular system (as suggested by [9]) with integration demands potentially exacerbating magnocellular-related temporal processing difficulties [17], and higher levels of internal noise could lead to difficulties with noise exclusion [23]. Critically, the motion-coherence task alone cannot distinguish between these explanations.

The case of motion-coherence difficulties in dyslexia is made more complex by the fact that difficulties with motion-coherence have not only been reported in dyslexia, but also in a range of other developmental conditions including autism [25,26], Fragile X Syndrome [27] and Williams Syndrome [28]. Therefore, it is not clear whether performance in motion-coherence tasks tells us anything specific about dyslexia, or whether it reflects a more general marker of atypical brain development (e.g. dorsal-stream vulnerability, [29]). Alternatively, it is possible that motion-coherence performance could be impaired in different neurodevelopmental conditions for different reasons. Resolving this question would inform fundamental debates about the causal role of atypical motion processing in these conditions and possible shared etiologies.

A further consideration relevant to this literature is heterogeneity among dyslexic individuals. Distinct subtypes of dyslexia have been proposed [30–32], but there is currently no consensus that motion-coherence processing is differentially affected in these subtypes (see [33,34]). Therefore, it seems appropriate to establish the reasons for atypical global motion processing in participants with a dyslexia diagnosis to understand overall group differences before developing and testing hypotheses regarding different subtypes.

In this study, we applied an approach that has not previously been used in dyslexic individuals to address the mechanisms of atypical global motion processing, while also enabling cross-syndrome comparisons. Specifically, we used an equivalent noise approach which quantifies local internal noise (the imprecision with which the directions of individual elements are estimated) and global integration ability (the extent to which direction information is pooled across elements). We used a motion-averaging task [35–37] in which there is no need to segregate signal dots from noise dots. Instead, the direction of dots on each trial was sampled from a Gaussian distribution, with stimulus noise (external noise) being manipulated by varying the standard deviation of this distribution. We also presented a motion-coherence task to investigate whether elevated motion-coherence thresholds are replicated in our sample of dyslexic children. Additionally, we presented static analogues of the averaging and coherence tasks using orientation information (see [37]) in order to investigate whether differences between dyslexic and non-dyslexic individuals are restricted to motion-processing tasks.

We recently used these motion and orientation tasks to give novel insights into autistic perception [36,37]. Despite reports of elevated motion-coherence thresholds in autistic populations in some (but not all) previous

studies (see [38], for meta-analysis), here we found no evidence of elevated motion-coherence thresholds in autistic children compared to typically developing children. There was also no evidence of increased levels of internal noise in autistic children. However, surprisingly, the autistic children demonstrated *enhanced* integration performance (i.e. increased sampling) in the motion-averaging task. These results led us to conclude that autistic children showed enhanced integration of motion signals, but did not show corresponding benefits in the motion-coherence task owing to the additional requirements to segregate signal from noise in this task. We also showed that these differences did not appear to generalize to static, orientation information [37]. By presenting the same tasks to a population of dyslexic children, we hoped to not only better understand visual perception in dyslexia, but also to compare patterns of performance in dyslexic and autistic children relative to typically developing children. If the pattern found in dyslexic children differed to that previously reported in autism, this would suggest that motion-processing differences are not a general marker of atypical development (e.g. [29]), but are instead condition-specific. Although previous studies have found elevated motion-coherence thresholds in both dyslexic and autistic populations ([10], but see also [39]), the current study helps us to determine whether the reasons for atypical global motion-processing are shared or distinct between the two conditions. Moreover, it allows us to compare the results with studies using similar tasks in adult clinical populations, such as individuals with migraine [40] and schizophrenia [41]. These results will therefore inform on the specificity of atypical global motion processing in dyslexia.

## 1.1. Hypotheses

(1) We hypothesized that dyslexic children would show elevated motion-coherence thresholds compared to typically developing children [11].

(2) We investigated whether dyslexic children show atypical sampling in the motion-averaging task compared to typically developing children. If dyslexic children show *decreased* sampling compared to typically developing children, it would be consistent with difficulties in integrating motion information [9]. If dyslexic children show *increased* sampling compared to typically developing children, it would suggest a similar pattern of performance as in autistic children [36,37].

(3) We hypothesized that dyslexic children would have higher estimates of internal noise obtained from the motion-averaging task, in line with empirical evidence of increased neural variability in dyslexia [20–22] and theoretical accounts [23].

(4) We hypothesized that dyslexic children would show similar orientation-coherence thresholds as typically developing children, following previous research finding no group differences in form-coherence tasks [8,16].

(5) We hypothesized that dyslexic children would show similar sampling estimates as typically developing children in the orientation-averaging task, as studies of global form processing have not previously revealed group differences [8,16]. Furthermore, it has been suggested that difficulties in dyslexic perception may be restricted to aspects of magnocellular or dorsal-stream function [14,29].

(6) We compared internal noise estimates in the orientation-averaging task between dyslexic children and typically developing children. If dyslexic children have higher estimates of internal noise in this task, it would be consistent with theories of increased neural noise in this condition [23]. However, if dyslexic children have similar levels of internal noise to typically developing children, it would suggest that internal noise relating to orientation discrimination (i.e. precision in estimating the orientation of individual elements) is not affected in dyslexia.

# 2. Methods

## 2.1. Pre-registration

The approved Stage 1 protocol is available here: https://osf.io/76w59/registrations.

## 2.2. Participants

A between-participants design was adopted where group membership (dyslexia, typically developing) was the independent variable under test. A power analysis was conducted using G*Power 3.1.9.4

**Table 1.** Demographics of participants included in the dataset. (Note: WIAT-III = Wechsler Individual Achievement Test, 3rd edition; TOWRE-2 = Test of Word Reading Efficiency, 2nd Edition; PDE = Phonemic Decoding Efficiency; SWE = Sight Word Efficiency.)

| | typically developing | dyslexic |
| --- | --- | --- |
| | M (s.d.) range | M (s.d.) range |
| age (years) | 10.87 (1.90) 8.19–14.91 | 11.75 (1.78) 8.30–14.97 |
| verbal IQ | 108.35 (9.28) 87–135 | 102.06 (9.32) 83–119 |
| performance IQ | 105.00 (12.38) 78–137 | 101.33 (14.70) 74–141 |
| full-scale IQ | 107.54 (10.30) 83–138 | 101.81 (11.58) 80–132 |
| WIAT-III spelling | 109.32 (13.74) 78–142[a] | 79.02 (8.49) 60–99 |
| TOWRE-2 PDE | 106.11 (10.53) 80–124[a] | 77.50 (6.78) 57–90 |
| reading and spelling composite | 107.71 (10.68) 92.5–133[a] | 78.26 (6.13) 58.5–88.5 |
| TOWRE-2 SWE | 102.05 (11.19) 85–128[a] | 79.83 (8.83) 55–100 |

[a]Note that these data were only present for 19 of the typically developing children.

software [42] in order to predict the sample size needed to detect group differences *a priori* in a two-tailed independent samples *t*-test. A meta-analysis of group differences between dyslexic and non-dyslexic samples in a motion-coherence task reported an effect size of $d = 0.747$ [11]. We used this effect size in the absence of studies using our averaging task in dyslexic samples. Our power analysis suggested that 48 participants per group were required to detect this effect size using a conservative, two-tailed test with 95% power and an alpha level of 0.05.

The final dataset (following exclusions) included 48 dyslexic children (26 female) and 48 typically developing children (25 female) aged between 8 and 14 years (table 1 for demographics). This relatively wide age range was chosen in order to facilitate reaching the required sample size. All participants included in the dataset were cognitively able (verbal IQ and performance IQ > 70, as measured by the Wechsler Abbreviated Scales of Intelligence, 2nd edition (WASI-2), [43]), had normal or corrected-to-normal vision (as assessed by a Snellen acuity chart; 20/20 or better) and wore their habitual optical correction during testing. Dyslexic children were only included in the dataset if they had a diagnosis of dyslexia (or were currently in the process of obtaining a dyslexia diagnosis, $n = 1$), and if they obtained a composite score of 89 or below on tests of non-word reading (Phonological Decoding Efficiency subtest of the Test of Word Reading Efficiency, 2nd edition (TOWRE-2) [44]) and spelling (Wechsler Individual Achievement Test, 3rd edition (WIAT-III) [45]). Note the cut-off score of 89 corresponds to 1.5 standard deviations below the mean of typically developing children in a similar study [46,47]. Eight additional children were excluded from the dyslexia group as their composite scores exceeded this cut-off. Children with a co-occurring autism diagnosis were excluded from the dyslexia group ($n = 1$) to enable cross-syndrome comparisons with the autistic children reported in Manning *et al.* [37]. No children with diagnosed developmental conditions were included in the typically developing group.

Prior to pre-registration, data had been collected from typically developing children within our target age range as part of a study by Manning *et al.* [37]. These data were used in the current dataset in order to minimize additional testing of children. Reading and spelling tests were not conducted with these participants, but parents were asked to report concerns about their child's reading or spelling in a background questionnaire and we only included typically developing children from the previous dataset if their parents reported no concerns about their reading or spelling. There were 29 existing datasets fitting our inclusion criteria that were used in the current dataset. We also collected new data from 19 typically developing children (following exclusions) to reach our final sample size of 48. For these newly recruited typically developing participants, we conducted reading and spelling tests and excluded them from the dataset if they obtained a composite score of 89 or below ($n = 0$).

We also excluded children from the dataset if their responses on the motion-processing tasks indicated a complete inability to perform the task. Specifically, participants were excluded if they (i) failed to achieve four consecutive correct responses in easy practice trials (see *Experimental Procedure*; $n = 1$ excluded), (ii) did not perform significantly above chance in catch trials (responding incorrectly to 4 or more of the 15 catch trials; $n = 1$), or (iii) did not complete all trials, in either

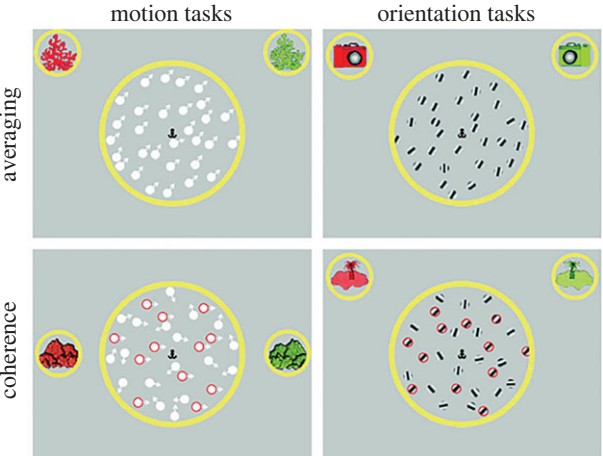

**Figure 1.** Stimuli in motion and orientation tasks. Schematic representation of stimuli used in the averaging and coherence tasks for motion and orientation information. Arrows and red circles are presented here—for illustrative purposes only—and indicate the direction of motion of dots and coherent (signal) elements, respectively. The upper panel depicts a trial from the high-noise condition for each averaging task, with the standard deviation of directions/orientations being 10°. The lower panel represents a trial from each coherence task, with 40% coherence in the direction/orientation of elements.

motion-processing task (motion-coherence or motion-averaging; $n = 1$). Additionally, participants who obtained a motion-coherence threshold estimate $\geq 100\%$ were excluded from the dataset ($n = 2$). Participants who failed to pass a criterion of four consecutive correct responses in easy practice trials ($n = 0$), performed significantly below chance on catch trials ($n = 1$) or did not complete the trials on either orientation task ($n = 3$), and participants who obtained an orientation-coherence threshold estimate $\geq 100\%$ ($n = 0$) were retained in the dataset but their data were only analysed for the motion-processing tasks. This resulted in a slightly smaller sample of 45 dyslexic and 47 typically developing children included in the analyses of orientation-processing tasks.

Participants were primarily recruited from schools and parent groups, and by re-contacting previous participants who agreed to be contacted about future studies (as in the previous study by [37]). When recruiting, we aimed to ensure close age-matching between typically developing and dyslexic children.

## 2.3. Apparatus and stimuli

Stimuli were presented on a Dell Precision M3800 laptop (2048 × 1152 pixels, 60 Hz) using MATLAB and elements of the Psychophysics Toolbox [48–50]. Stimuli were presented within a yellow-bordered circular aperture (15° diameter) in the centre of a grey screen, with an anchor-shaped fixation point (0.57° × 0.57°), which remained on the screen throughout the trials (figure 1). For the motion tasks, stimuli were comprised 100 white dots (0.44° diameter) displaced by 0.075° every three frames, yielding a dot speed of $1.5° \text{ s}^{-1}$. In the orientation tasks, stimuli were 100 static Gabor patches with a spatial frequency of 3.4 c deg$^{-1}$, presented with random phases and at 50% contrast, within circular (hard-edged) apertures (diameter 0.44°). Red and green images were presented in small yellow-bordered circular apertures to the left and right of the central aperture containing the stimuli. These images served as reference points.

## 2.4. Experimental procedure

Averaging and coherence tasks were presented for both motion direction and static orientation information, as in Manning *et al.* [37]. The averaging tasks consisted of 'no-noise' and 'high-noise' conditions. In the no-noise condition, the standard deviation of directions or orientations was fixed at 0° while the mean direction or orientation was varied. In the high-noise condition, the mean offset was fixed (±45° in the motion-averaging task, and ±22.5° in the orientation-averaging task), while the standard deviation of directions or orientations was varied. In the coherence tasks, a proportion of elements were designated 'signal' elements with a coherent direction (±90°) or orientation (±45°), while the remaining elements were 'noise' elements with random directions or orientations. The proportion of signal elements to noise elements was varied. The tasks were presented within the

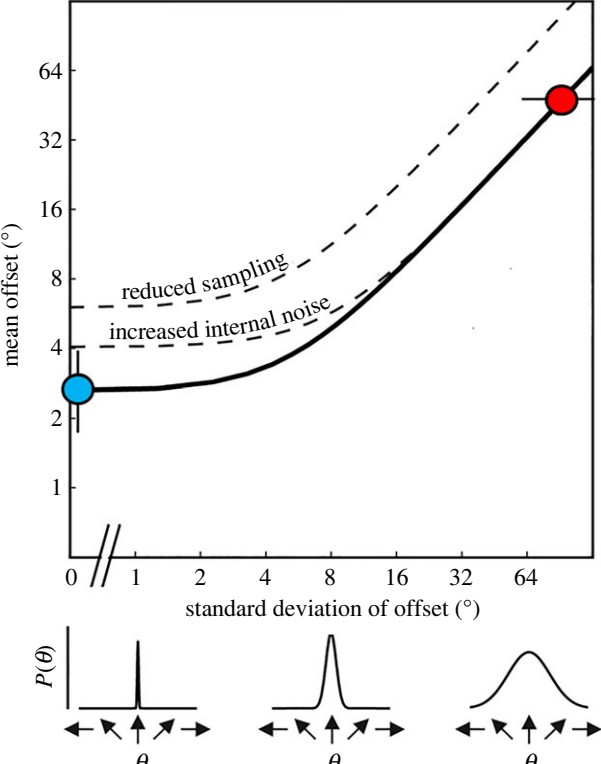

**Figure 2.** Equivalent noise function. Diagram of example equivalent noise function constrained by data from no-noise (blue) and high-noise (red) conditions of the averaging tasks. The mean direction/orientation offset (left or right of vertical) that is required to discriminate at a certain level of accuracy (i.e. threshold) increases as a function of the standard deviation of the distribution of directions/orientations from which the elements are drawn (external noise). Thresholds are relatively unaffected by low levels of external noise, as internal noise dominates. However, as external noise is increased further, the internal noise is swamped and thresholds start to increase. In the no-noise condition (blue), the standard deviation is fixed at 0° and the no-noise threshold is obtained by varying the directional/orientation offset. In the high-noise condition (red), the mean offset is fixed at ±45° in the motion task (as depicted here), and ±22.5° in the orientation task, and the standard deviation is varied to find the maximum tolerable noise. Reduced sampling shifts the function upwards, with reduced discrimination performance at all levels of internal noise. By contrast, increased levels of internal noise lead to higher thresholds at low levels of external noise and a rightwards shift of the elbow of the function, so that more external noise is required before thresholds start to increase.

context of child-friendly games, in which children were asked which direction the 'fish' were swimming in (for the motion tasks), and which way the 'jellyfish with stripes on their back' were facing (for the orientation tasks). Trials were initiated by a keypress from the experimenter. Children responded by pointing or responding 'red' or 'green' using the reference points (figure 2), following which the experimenter relayed their responses to the computer using a keyboard.

For each of the four tasks (motion-averaging, motion-coherence, orientation-averaging and orientation-coherence), children first completed a demonstration and criterion phase (level 1) to familiarize them with the tasks. Children were first presented with four demonstration trials and were then presented with up to 20 'easy' criterion trials until they reached a criterion of four consecutive correct responses. In the averaging tasks, these trials had a standard deviation of 0° (no-noise) and a mean direction of ±45° or mean orientation of ±22.5°. In the coherence tasks, the trials had 100% coherence. Visual and verbal feedbacks were provided. Next, children completed a practice phase (level 2) in which they completed eight trials of increasing difficulty. Finally, children completed a threshold estimation phase (level 3) in which stimulus difficulty (mean offset, standard deviation or proportion coherence) was controlled by QUEST [51]. The no-noise and high-noise conditions of the averaging tasks each consisted of a QUEST staircase of 75 trials, which were randomly interleaved. The coherence tasks consisted of a single QUEST staircase of 75 trials. Each QUEST generated values in log units, with starting points corresponding to a mean offset of ±2.5° in the no-noise condition of the averaging tasks, a standard deviation of 0.001° in the high-noise condition of the averaging tasks and 50% coherence in the coherence tasks. Each QUEST had a prior with a standard deviation of 2, a slope (beta) value of 3.5 specifying the steepness of the Weibull function and a lapse rate set to 0.01. An additional 15 catch trials (using the same stimuli used

in the criterion trials) were randomly interleaved into each task. Therefore, there were 165 trials in each averaging task and 90 trials in each coherence task. These trials were divided into four blocks for each task. No feedback was given during the threshold estimation phase, but the child did receive general encouragement throughout testing. In order to keep children attentive and motivated, after each block, they were shown a graph showing 'points' that they and a cartoon 'opponent' had attained. These values were randomly jittered around a fixed set of values which ensured that participants always received slightly more points than the opponent.

## 2.5. General procedure

The study received ethical approval by the Medical Sciences Interdivisional Research Ethics Committee (IDREC) at University of Oxford (R45641/RE001). Parents gave written informed consent and children gave written assent. Children were seen individually in a dimly-lit, quiet area at the University. Children completed coherence and averaging tasks for both motion and orientation stimuli, seated at a viewing distance of 51 cm from the computer screen with a chin-rest. The experimenter monitored children's fixation during the tasks and regularly reminded children to fixate, only initiating trials when children were seen to be attending. The motion and orientation tasks were completed in separate sessions each lasting approximately 25 min, with the order of sessions being counterbalanced between participants. Within each session, the order of tasks (coherence, averaging) was also counterbalanced between participants. Children used a record card and a stamper to record their progress through the 'levels' of the 'games'.

The WASI-II, the TOWRE-2 (sight word reading efficiency and phonological decoding efficiency subtests), the WIAT-III spelling subtest and an acuity test were completed in further sessions. If children had completed the WASI-II or WIAT-III in the last 3 years as part of a previous study [52,53], we did not rerun these tests in order to avoid practice effects and 'over-testing' children. Rather we used the previously collected scores. New TOWRE-2 scores were collected using an alternate form than used in the previous study. All tasks were completed in approximately 1.5 h.

## 2.6. COVID-19 measures

Data were collected in October 2020 and between April 2021 and September 2021, at which time, COVID-19 control measures were mandated by the institution. These measures, which were put in place after in-principle acceptance was granted, included requiring participating children aged 11 years and above to wear masks (unless exempt) and for researchers to wear Level 1 PPE (masks, visor, gloves and aprons).

## 2.7. Analysis

### 2.7.1. Threshold measures

Three thresholds were computed for the motion and orientation tasks: (i) the angular direction or orientation offset leading to 84% correct performance in the no-noise condition of the averaging task (*no-noise threshold*), (ii) the standard deviation leading to 84% correct performance in the high-noise conditions of the averaging task (*maximum tolerable noise* (MTN)), and (iii) the proportion coherence leading to 84% correct performance in the coherence task (*coherence threshold*). Thresholds were taken as the mean of QUEST's posterior probability density function.

### 2.7.2. Equivalent noise analysis

We used equivalent noise analysis to estimate two limits on integration performance in the averaging tasks: internal noise ($\sigma_{int}^2$), quantifying imprecision in the estimation of the directions/orientations of individual elements, and sampling ($n_{samp}$), quantifying the effective number of elements that are averaged. The logic behind equivalent noise analysis is that discrimination thresholds ($\sigma_{obs}^2$) are limited by both internal noise and external noise ($\sigma_{ext}^2$) in the stimulus, and that internal noise levels can be estimated by relating discrimination thresholds to external noise (i.e. the standard deviation of directions/orientations in the stimuli in our averaging tasks, figure 2). Specifically:

$$\sigma_{obs}^2 = \frac{[\sigma_{int}^2 + \sigma_{ext}^2]}{n_{samp}}.$$

We used the efficient version of the equivalent noise model designed for use with children and clinical populations [36,37,40,41]. This version of the equivalent noise model uses two highly informative points on the equivalent noise function to constrain its fit, corresponding to the no-noise and high-noise conditions of our averaging tasks (figure 2). Sampling ($n_{samp}$) was estimated by transforming the MTN in the high-noise condition:

$$n_{samp} = \exp(A \times MTN^2 + B \times MTN + C),$$

where $A$, $B$ and $C$ were 0.0001, 0.0357 and −1.8093 for the motion task, and 0.0006, 0.0652 and −1.6843 for the orientation task, respectively (based on best-fitting values from Monte Carlo simulations, see [40]). Internal noise was then estimated by rearranging the first equation with external noise ($\sigma^2_{ext}$) set to 0:

$$\sigma^2_{int} = \sigma^2_{obs} \times n_{samp}.$$

## 2.8. Statistical analysis

For each of our six hypotheses, we planned to conduct a two-tailed independent samples $t$-test comparing dyslexic and typically developing children using the relevant measure (coherence threshold, sampling and internal noise) and an alpha level of 0.05 (table 2). We planned to supplement our analyses with Bayesian $t$-tests with default Cauchy priors with a distribution of (0,1) using JASP software ([54]; see also [37]). These Bayesian analyses allow us to quantify the evidence for the null hypothesis in the event that the $t$-test results are non-significant ($p > 0.05$). Bayes factors ($BF_{10}$) between 1/3 and 3 are interpreted as inconclusive results [55]. In the event that the data are not normally distributed (as assessed using the Shapiro–Wilk test), we planned to use non-parametric equivalents (Mann–Whitney $U$-test and Bayesian Mann–Whitney $U$-test). Our hypotheses were tested in the full sample, but we also conducted exploratory, follow-up analyses with only the participants for whom we have reading and spelling scores to investigate whether the same pattern of results is seen. When interpreting these exploratory results we consider the effects of reduced power.

# 3. Results

## 3.1. Catch trial performance and normality of dependent variables

Lapse rates (i.e. the proportion of errors made on 'easy' catch trials) were low for all tasks, in both groups (table 3). Estimates for no-noise thresholds, MTN thresholds, coherence thresholds, internal noise and sampling for each individual are shown in figures 3 and 4, for the motion and orientation tasks, respectively. Shapiro–Wilk tests showed that all dependent variables related to our pre-registered hypotheses (coherence thresholds, internal noise and sampling) deviated significantly from normality in either one or both groups (see the electronic supplementary material, table S1). We therefore tested our hypotheses using the Mann–Whitney $U$-test and its Bayesian equivalent.

## 3.2. Comparing dyslexic and typically developing children's motion-processing performance

The dyslexic children had slightly higher motion-coherence thresholds than the typically developing children, as shown in figure 3$c$, but the group difference was not significant ($W = 904$, $p = 0.07$, rank-biserial correlation = −0.22), with the Bayes factor being very close to 1 ($BF_{10} = 0.98$), suggesting inconclusive, equivocal evidence for either the null or alternative hypothesis of group differences in motion-coherence thresholds (H1).

Figure 3$d$ shows that the range of sampling estimates in the motion-averaging task was wider in the typically developing group than in the dyslexic group, with some typically developing children effectively sampling more dots than the dyslexic group. However, the median values across the groups were similar, with no significant group difference ($W = 1281$, $p = 0.35$, rank-biserial correlation = 0.11). The Bayes factor fell in the inconclusive range ($BF_{10} = 0.37$), suggesting relatively more evidence for the null hypothesis (no group differences) than the alternative hypothesis of group differences in sampling (H2), but with inconclusive evidence.

The dyslexic children exhibited slightly higher internal noise estimates in the motion-averaging task than the typically developing children (figure 3$e$). This difference was significant on the Mann–Whitney

**Table 2.** Pre-registered research questions and hypotheses.

| question | hypothesis | sampling plan | analysis plan | interpretation given different outcomes |
|---|---|---|---|---|
| 1. do dyslexic children differ from typically developing children in motion-coherence thresholds? | dyslexic children will have higher motion-coherence thresholds than typically developing children | 48 participants per group. Power analysis based on d = 0.747, 95% power, $\alpha = 0.05$ | two-tailed independent samples t-test and Bayesian t-test on motion-coherence thresholds | (a) if dyslexic children have higher motion-coherence thresholds than typically developing children ($p < 0.05$), this will be interpreted as reduced sensitivity in dyslexia in this task, consistent with previous research. (b) If $p > 0.05$ and $BF_{10} < 1/3$ (i.e. evidence in support of the null hypothesis), this suggests that the groups do not differ in motion-coherence sensitivity. (c) If $p > 0.05$ and $BF_{10} > 1/3$, this suggests that the data are inconclusive. (d) If dyslexic children have lower motion-coherence thresholds than typically developing children ($p < 0.05$), this will be interpreted as increased sensitivity in this task, in contrast with previous research |

(Continued.)

**Table 2.** (Continued.)

| question | hypothesis | sampling plan | analysis plan | interpretation given different outcomes |
|---|---|---|---|---|
| 2. do dyslexic children differ from typically developing children in sampling estimates in the motion-averaging task? | dyslexic children will differ from typically developing children in sampling estimates in the motion-averaging task | 48 participants per group. Power analysis based on d = 0.747, 95% power, $\alpha = 0.05$ | two-tailed independent samples $t$-test and Bayesian $t$-test on sampling estimates resulting from equivalent noise analysis of performance in the motion-averaging task | (a) if dyslexic children have lower sampling estimates than typically developing children ($p < 0.05$), this will be interpreted as reduced motion integration. (b) If $p > 0.05$ and $BF_{10} < 1/3$, this suggests that the groups do not differ in motion integration abilities. (c) If $p > 0.05$ and $BF_{10} > 1/3$, this suggests that the data are inconclusive. (d) If dyslexic children have higher sampling estimates than typically developing children ($p < 0.05$), this will be interpreted as increased motion integration: a similar pattern found in autism research |

(Continued.)

**Table 2.** (*Continued.*)

| question | hypothesis | sampling plan | analysis plan | interpretation given different outcomes |
|---|---|---|---|---|
| 3. do dyslexic children differ from typically developing children in internal noise estimates in the motion-averaging task? | dyslexic children will have higher estimates of internal noise obtained from the motion-averaging task than typically developing children | 48 participants per group. Power analysis based on $d = 0.747$, 95% power, $\alpha = 0.05$ | two-tailed independent samples $t$-test and Bayesian $t$-test on internal noise estimates resulting from equivalent noise analysis of performance in the motion-averaging task | (a) if dyslexic children have higher internal noise estimates than typically developing children ($p < 0.05$), this will be interpreted as reduced precision of estimating local dot directions in dyslexia. (b) If $p > 0.05$ and $BF_{10} < 1/3$, this suggests that the groups do not differ in terms of precision of estimating local dot directions. (c) If $p > 0.05$ and $BF_{10} > 1/3$, this suggests that the data are inconclusive. (d) If dyslexic children have lower internal noise estimates than typically developing children ($p < 0.05$), this will be interpreted as increased precision of estimating local dot directions in dyslexia |

(*Continued.*)

**Table 2.** (*Continued.*)

| question | hypothesis | sampling plan | analysis plan | interpretation given different outcomes |
|---|---|---|---|---|
| 4. do dyslexic children differ from typically developing children in orientation-coherence thresholds? | dyslexic children will not differ from typically developing children in orientation-coherence thresholds | 48 participants per group based on the power analysis for the motion tasks, but participants will be excluded from the orientation task analysis if they are unable to perform the orientation tasks | two-tailed independent samples *t*-test and Bayesian *t*-test on orientation-coherence thresholds | (a) if dyslexic children have higher orientation-coherence thresholds than typically developing children ($p < 0.05$), this will be interpreted as reduced sensitivity in dyslexia in this task. (b) If $p > 0.05$ and $BF_{10} < 1/3$, this suggests that the groups do not differ in orientation-coherence sensitivity. (c) If $p > 0.05$ and $BF_{10} > 1/3$, this suggests that the data are inconclusive. (d) If dyslexic children have lower orientation-coherence thresholds than typically developing children ($p < 0.05$) this will be interpreted as increased sensitivity in this task |

(*Continued.*)

**Table 2.** (*Continued.*)

| question | hypothesis | sampling plan | analysis plan | interpretation given different outcomes |
|---|---|---|---|---|
| 5. do dyslexic children differ from typically developing children in sampling estimates in the orientation-averaging task? | dyslexic children will not differ from typically developing children in sampling estimates in the orientation-averaging task | 48 participants per group based on the power analysis for the motion tasks, but participants will be excluded from the orientation task analysis if they are unable to perform the orientation tasks | two-tailed independent samples $t$-test and Bayesian $t$-test on sampling estimates resulting from equivalent noise analysis of performance in the orientation-averaging task | (a) if dyslexic children have lower sampling estimates than typically developing children ($p < 0.05$), this will be interpreted as reduced orientation integration. (b) If $p > 0.05$ and $BF_{10} < 1/3$, this suggests that the groups do not differ in orientation integration abilities. (c) If $p > 0.05$ and $BF_{10} > 1/3$, this suggests that the data are inconclusive. (d) If dyslexic children have higher sampling estimates than typically developing children ($p < 0.05$), this will be interpreted as increased motion integration: a similar pattern found in autism research |

(*Continued.*)

**Table 2.** (Continued.)

| question | hypothesis | sampling plan | analysis plan | interpretation given different outcomes |
|---|---|---|---|---|
| 6. do dyslexic children differ from typically developing children in internal noise estimates in the orientation-averaging task? | dyslexic children will either have higher or similar internal noise estimates compared to typically developing children | 48 participants per group based on the power analysis for the motion tasks, but participants will be excluded from the orientation task analysis if they are unable to perform the orientation tasks | two-tailed independent samples $t$-test and Bayesian $t$-test on internal noise estimates resulting from equivalent noise analysis of performance in the orientation-averaging task | (a) if dyslexic children have higher internal noise estimates than typically developing children ($p < 0.05$), this will be interpreted as reduced precision of estimating local orientations in dyslexia, consistent with theories of increased internal noise in dyslexia. (b) If $p > 0.05$ and $BF_{10} < 1/3$, this suggests that the groups do not differ in terms of precision of estimating local orientations. (c) If $p > 0.05$ and $BF_{10} > 1/3$, this suggests that the data are inconclusive. (d) If dyslexic children have lower internal noise estimates than typically developing children ($p < 0.05$), this will be interpreted as increased precision of estimating local orientations in dyslexia |

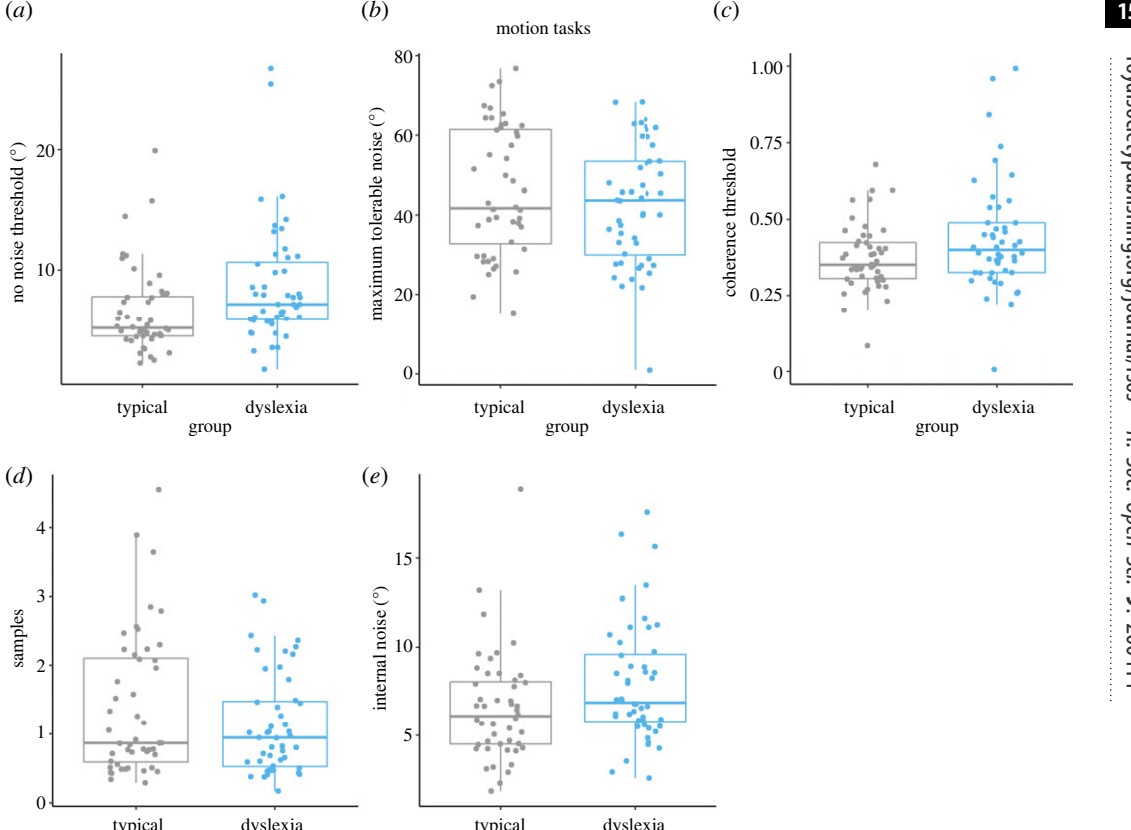

**Figure 3.** Box plots showing individual threshold estimates (*a–c*) and equivalent noise modelling estimates (*d,e*) in the motion tasks, for typically developing children (grey) and dyslexic children (blue). Group comparisons were conducted on coherence thresholds (*c*), sampling estimates (*d*) and internal noise estimates (*e*).

**Table 3.** Mean, standard deviation and range of proportion of errors made in catch trials for typically developing children and dyslexic children, in each task.

| task | typically developing M (s.d.) range | dyslexic M (s.d.) range |
|---|---|---|
| motion-averaging | 0.013 (0.026) 0–0.067 | 0.015 (0.042) 0–0.200 |
| motion-coherence | 0.006 (0.019) 0–0.067 | 0.017 (0.035) 0–0.133 |
| orientation-averaging | 0.013 (0.033) 0–0.133 | 0.027 (0.039) 0–0.133 |
| orientation-coherence | 0.011 (0.025) 0–0.067 | 0.016 (0.035) 0–0.133 |

*U*-test ($W = 852$, $p = 0.03$, rank-biserial correlation $= -0.26$), in support of the alternative hypothesis. The Bayes factor suggested relatively more evidence for the alternative hypothesis of group differences in internal noise (H3) than for the null hypothesis ($BF_{10} = 2.00$), i.e. the data were twice as likely under the alternative hypothesis than the null hypothesis. However, this evidence was still in the weak or inconclusive range.

## 3.3. Comparing dyslexic and typically developing children's orientation-processing performance

The dyslexic children had slightly higher orientation-coherence thresholds than typically developing children (figure 4*c*): a difference that was significant on the Mann–Whitney *U*-test ($W = 730$, $p = 0.01$, rank-biserial correlation $= -0.31$). While the Bayes factor supported relatively more evidence for the

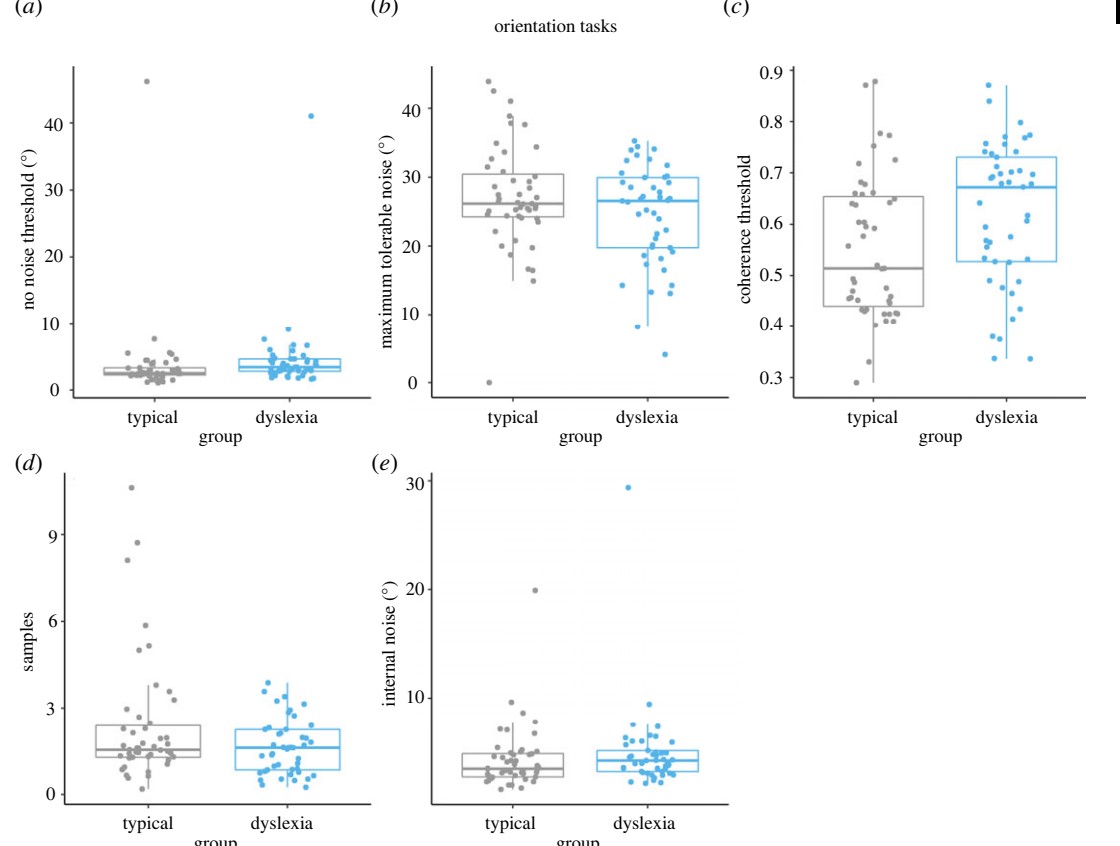

**Figure 4.** Box plots showing individual threshold estimates (*a–c*) and equivalent noise modelling estimates (*d,e*) in the orientation tasks, for typically developing children (grey) and dyslexic children (blue). Group comparisons were conducted on coherence thresholds (*c*), sampling estimates (*d*) and internal noise estimates (*e*).

alternative hypothesis of group differences in orientation-coherence thresholds (H4) than the null hypothesis ($BF_{10} = 1.80$), it fell within the inconclusive range.

For the orientation-averaging task (like the motion-averaging task), the range of sampling estimates for the typically developing group exceeded that for the dyslexic group, with some children in the typically developing group having higher sampling estimates than those in the dyslexic group (figure 4*d*). However, the dyslexic group did not differ significantly from the typically developing group in sampling estimates ($W = 1208$, $p = 0.24$, rank-biserial correlation = 0.14). The corresponding Bayes factor showed relatively more evidence for the null hypothesis than the alternative hypothesis (H5), but the evidence was inconclusive ($BF_{10} = 0.60$)

Estimates of internal noise were relatively similar across dyslexic and typically developing groups in the orientation-averaging task, and there was an outlying point in each group (figure 4*e*). The results of the statistical tests confirmed that the dyslexic group did not differ significantly from typically developing children in terms of internal noise estimates ($W = 874$, $p = 0.15$, rank-biserial correlation = $-0.17$), and the Bayes factors showed relatively more evidence for the null hypothesis, although the evidence was inconclusive ($BF_{10} = 0.45$).

## 3.4. Exploratory analyses: subsample analysis

As we did not have TOWRE-2 and WIAT-III Spelling scores for the typically developing participants who were tested as part of our previous study [37], we re-ran the analyses excluding these participants, to determine whether it changed the pattern of results. In this smaller sample (48 dyslexic children and 19 typically developing children), none of the group comparisons were significant (all $p \geq 0.13$). The lack of group differences in motion-coherence thresholds and in sampling estimates in the motion-averaging task were conclusive in this sample ($BF_{10} = 0.32$ and $BF_{10} = 0.30$, respectively), while all other Bayes factors were in the inconclusive range, between 1/3 and 3.

## 3.5. Exploratory analyses: controlling for age

While we aimed to ensure the groups were well-matched in age, table 1 shows that the dyslexic children were slightly older than the typically developing children (Cohen's $d = 0.47$). As the measures under test have been shown to change with age [56], we conducted exploratory analyses where we assessed group differences while controlling for age, using a non-parametric ANCOVA (Quade's ANCOVA; [57,58]) in $R$, for each of the hypotheses under test. This procedure involved converting the dependent variable and covariate (age) to ranks, and then to deviation scores (by subtracting the mean rank from each datapoint). Next, the regression coefficient was calculated and the deviation rank of the dependent variable was predicted from the deviation rank of the covariate. The predicted dependent variable deviation rank was then subtracted from the observed deviation rank, yielding a residual deviation rank score, which we then used to test for group differences using a parametric test. We also conducted a Bayesian $t$-test on the residual deviation rank scores, in JASP. The results are presented in the electronic supplementary material, table S2.

The results are largely in line with the pre-registered analyses, where we found significant group differences ($p < 0.05$) in motion internal noise and orientation-coherence threshold estimates. However, once we had controlled for age, there was also a marginally significant group difference in motion-coherence thresholds ($p = 0.047$), with higher thresholds in the dyslexic children. However, the Bayesian analyses which controlled for age suggest that group differences in motion-coherence thresholds and motion internal noise are still within the inconclusive range, with orientation-coherence thresholds being the only measure to show strong evidence for group differences, once age is controlled for (with dyslexic children having higher orientation-coherence thresholds, $BF_{10} = 13.66$). Scatterplots showing relationships between age and the dependent variables for each group are provided in the electronic supplementary material, figure S1.

## 3.6. Exploratory analyses: correlations between motion-coherence thresholds and internal noise and sampling

In typical development, motion-coherence thresholds have been shown to be related to sampling, not internal noise [56]. To determine whether the same pattern is found in the dyslexic children and typically developing children in the current sample, we conducted correlations between coherence thresholds and sampling and internal noise, while controlling for age, for both motion and orientation tasks. The same pattern was found for both groups in both tasks, with coherence thresholds being significantly related to sampling but not internal noise (electronic supplementary material, table S3).

# 4. Discussion

In this study, dyslexic and typically developing children judged motion direction and orientation and their performance was compared on three key measures: coherence thresholds, internal noise and sampling. Our pre-registered analyses showed that dyslexic children had significantly higher internal noise estimates in the motion-averaging task and significantly higher coherence thresholds in the orientation task, compared to typically developing children, although these group differences were subtle. All other group differences were non-significant. We supplemented frequentist statistics with Bayesian statistics, to quantify the relative evidence for the null and alternative hypotheses to make inferences when non-significant differences were obtained. This analysis revealed inconclusive evidence for or against the hypothesized group differences for the measures in which non-significant results were obtained. It is also worth noting that the Bayesian analyses did not suggest strong, conclusive evidence for group differences when comparing internal noise estimates in the motion task and orientation-coherence thresholds, with the data being only approximately two times more likely under the alternative hypothesis than the null hypothesis, in each case.

Since the sample of dyslexic children was slightly older than our typically developing children, we also conducted exploratory analyses in which we controlled for age. The pattern of results was similar, though the inter-group difference in motion-coherence thresholds was now significant, with the dyslexic children having significantly higher motion-coherence thresholds on average than typically developing children. Additionally, the Bayesian analyses showed that the evidence for group differences in orientation-coherence thresholds was now much stronger once controlling for age.

With respect to the results from the motion tasks, we hypothesized (H1) that dyslexic children would show higher motion-coherence thresholds than typically developing children, following previous research [11]. Indeed, the dyslexic children did, on average, have slightly higher motion-coherence thresholds, though this difference was only significant when age was controlled for. It has previously been shown that motion-coherence thresholds vary as a function of age (e.g. [56]), which could explain why these group differences emerged only once controlling for age. However, it is also important to note that the group differences were subtle, with considerable overlap in motion-coherence thresholds between children with and without dyslexia. Therefore, future research is needed to understand the importance of different stimulus and task parameters, to see if this affects the extent of group differences.

Interestingly, we found no evidence that dyslexic children differed from typically developing children in the effective number of samples that they averaged over in the motion-averaging task (H2). Rather, dyslexic children had higher internal noise estimates than typically developing children (H3). This suggests that dyslexic children are limited primarily by local limits to motion integration rather than global limits. Specifically, dyslexic children appear to be less precise at estimating the direction of each individual dot than typically developing children. This finding is consistent with dyslexia being associated with increased neural noise [23], in this case *additive noise*. As internal noise estimates were not significantly related to motion-coherence thresholds in either group, it could be that sensitivity in motion-coherence tasks is limited primarily by noise exclusion differences in dyslexia [18,19]. Future research could also investigate whether the differences in internal noise and motion-coherence sensitivity reported here are related to increased visual crowding in dyslexia [59,60], whereby the multiple dots in the display interfere with the perception of each individual dot's motion.

By presenting corresponding orientation tasks, we aimed to determine the domain-generality of perceptual differences in dyslexia. In the orientation task, the only significant group difference was for orientation-coherence thresholds (H4), with no evidence of group differences in either sampling (H5) or internal noise (H6) estimates. We had not expected to find a group difference in orientation-coherence thresholds, following previous results using form-coherence tasks [8,16]. However, the previous tasks were quite different from our own, requiring participants to detect a coherent circle in line segments, rather than discriminating the overall orientation. Additionally, the previous studies used smaller samples that may have not been powered to detect significant group differences. Consistent with this possibility, threshold plots in Hansen *et al.* [8] and Conlon *et al.* [16] (figures 2 and 3, respectively), suggest that the form-coherence thresholds were in fact higher in the dyslexic group than the typically developing group, even if these differences did not reach statistical significance. It is possible that orientation-coherence thresholds were elevated in dyslexia, despite no group differences in internal noise or sampling, because the orientation-coherence task additionally requires noise exclusion, which might be reduced in dyslexic children [18,19].

When considering the pattern of results *across* motion and orientation tasks, it seems that perceptual differences in dyslexia are not restricted to motion-processing tasks designed to tax the magnocellular or dorsal stream [14,29]. Instead, when age was controlled for in exploratory analyses, both orientation- and motion-coherence thresholds were significantly elevated in dyslexic children. However, it seems that orientation- and motion-processing might be affected differently in dyslexia, as there was no evidence that internal noise was increased in dyslexia for the orientation tasks, as it was for the motion tasks. Therefore, it is important to consider that increases in neural noise in dyslexia might be task-specific. However, given the relatively inconclusive nature of our results, further research with larger samples is required to confirm this possibility.

More generally, we note that Bayesian statistics did not show compelling evidence to support or refute any of our hypotheses (apart from in the exploratory analysis of orientation-coherence thresholds, when age was controlled for). Previous studies into perceptual processing in dyslexia have seldom used Bayesian statistics, so that it is difficult to directly compare the strength of evidence with these previous studies. However, when looking at the variability in performance, it is clear to see why the results only constitute weak or inconclusive evidence, as the groups are highly overlapping in performance and there is considerable interindividual variability within each group. Therefore, not all dyslexic children show differences in visual processing relative to the typically developing comparison group. Indeed, this point has been made before in relation to performance on visual tasks (e.g. [61,–63]). As a result, it is unlikely that atypical visual processing could be used diagnostically to distinguish between those with and without dyslexia.

What could explain this variability? It has been proposed that there are different subtypes of dyslexia (e.g. phonological and surface dyslexia; see [30–32]), and it is possible that these subtypes relate to the

nature and extent of differences in the visual processing of children with dyslexia. We decided to analyse performance for the whole sample of dyslexic children, because we wanted to maximize statistical power, but also because there is a lack of strong evidence for the existence of subtypes with clearly distinct cognitive or biological profiles [5]. Previous studies of motion perception have tended not to break the dyslexia sample into subgroups, and one study that did break their group up reported elevated motion-coherence thresholds across all subgroups [33]. While future studies could investigate whether different subtypes of dyslexia are affected differently, we note that considerable variability was also found in the typically developing sample, and there were no clearly defined clusters in the dyslexia group. Therefore, it might be that it is more informative to study the effect of continuous dimensions on performance than to subdivide the dyslexia sample into smaller groups. Exactly which continuous dimensions are relevant to performance remains to be tested, but attentional abilities could be one candidate to investigate in future research, especially as dyslexia commonly co-occurs with attention-deficit/hyperactivity disorder [64,65]. The cumulative risk-resilience model [66] proposes that multiple factors increase the risk of dyslexia. Therefore, examining visual processing alongside other dimensions could support our understanding of the role of visual processing in dyslexia. Furthermore, dissociable aspects of reading skill could be related to variability in task performance [67].

There are four limitations of our study that need to be considered. First, we did not have reading and spelling data for all typically developing children, as we reused the data from some children who had participated in a previous study. While none of the parents of these typically developing children reported literacy concerns for their children, it could have been that there were some children in the typically developing group who had reading and spelling abilities consistent with a dyslexia diagnosis, thus minimizing potential group differences in task performance. However, this is unlikely because exploratory analysis on the subset of participants for whom we had reading and spelling scores showed no significant group differences in any measures. It is difficult to conclude much from these exploratory analyses, however, because the subset had reduced power and unequal group sizes. A second limitation of this study is that the sample covered a reasonably wide age range. While this facilitated the collection of enough data to achieve the sample size determined by our *a priori* power calculation, it is possible that there could be different developmental trajectories in dyslexia and typical development—a possibility which could be explored in future research. The wide age range could potentially have obscured group differences, although Benassi *et al.*'s [11] meta-analysis (which we based our power analysis on) included studies which used similarly wide age ranges and found large group differences in motion-coherence thresholds (e.g. [19,68]). A third limitation is that the new data were collected during the COVID-19 pandemic, and it is possible that this may have influenced the population willing to participate. A fourth limitation is that we did not collect response time data, because responses were relayed to the computer by the researcher. While previous studies using the equivalent noise approach (e.g. [36,41]) and previous studies of motion processing in dyslexia (e.g. [8]) have also relied purely on accuracy indices, group differences between dyslexic children and typically developing children have recently been shown in drift-diffusion model studies, which model both accuracy and response time data [52,69]. These studies have shown that dyslexic children accumulate evidence from motion displays more slowly than typically developing children. By combining both accuracy and response time, it might be possible to obtain more sensitive estimates of children's perceptual performance [70], while also accounting for speed-accuracy trade-offs.

Notwithstanding these limitations, it is interesting to compare the pattern of results obtained here with those reported in previous studies comparing autistic and typically developing children using the same paradigm [36,37]. Autistic children showed no significant differences in internal noise or coherence thresholds in either task, but showed *increased* sampling in the motion tasks, compared to typically developing children. The pattern of differences found here for dyslexic children compared to typically developing children (increased internal noise in the motion task and elevated orientation-coherence thresholds) is therefore *different* to that reported for autistic children. Therefore, we suggest that perceptual differences are specific to each condition (see also [52,71]. Accounts which suggest general impairments in motion processing across neurodevelopmental disorders (e.g. dorsal-stream vulnerability, [29]) need to be updated to reflect these nuances, and future work is needed to understand why different aspects of visual processing are affected in different developmental conditions.

Ethics. The study received ethical approval by the Medical Sciences Interdivisional Research Ethics Committee (IDREC) at the University of Oxford (R45641/RE001).
Data accessibility. Parents of new participants were asked to consent to anonymized data being made publicly available, so that their data could be shared. We also obtained ethical approval to share the data from typically developing

children previously collected in the earlier study [37]. The data, experimental code and analysis code are available at: https://doi.org/10.17605/OSF.IO/76W59 [72]. Material is also provided in the electronic supplementary material [73].

Authors' contributions. C.M.: conceptualization, data curation, formal analysis, funding acquisition, investigation, project administration, software, supervision, visualization and writing—original draft; V.H.: data curation, investigation, project administration and writing—review and editing; M.S.T.: methodology, software and writing—review and editing; S.C.D.: methodology, software and writing—review and editing.

All authors gave final approval for publication and agreed to be held accountable for the work performed therein.

Conflict of interest declaration. We declare we have no competing interests.

Funding. This study was funded by a University of Oxford Returning Carer's Grant and a Sir Henry Wellcome Postdoctoral Fellowship (grant no. 204685/Z/16/Z) awarded to C.M.

Acknowledgements. We are grateful to Heather Woods and Katie Hurman for help with data collection. We also thank the families who participated, along with schools, dyslexia organizations, Irina Lepadatu and the Oxford Babylab for help with participant recruitment.

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
