## [Peer Review File · Royal Society Open Science]

Review History

RSOS-200414.R0 (Original submission)

Review form: Reviewer 1 (Jiawei Zhou)

Do you have any ethical concerns with this paper?

No

Recommendation?

Accept with minor revision

Comments to the Author(s)

In this report, Manning et. al. applied a research proposal using averaging and coherence tasks with motion direction or static orientation information to address the mechanisms of atypical global motion processing in dyslexic individuals. In my view, this will be an interesting study. Since the tasks provided here have already been applied in autistic individuals, I just have the following minor comments that I would like the authors to address:

1. For the including of dyslexic participants, are there any Guidelines on the treatment and diagnosis of dyslexia that author refers to? Please add.
2. During the experiment, is there any method to fix children's head and/or maintain their fixation? If not, will the uncertain of fixation or head movement add additional noise?
3. In statistical analysis, independent samples t-test would also be added to ensure the two groups are age-matched.

Review form: Reviewer 2 (Nic Badcock)

Do you have any ethical concerns with this paper?

No

Recommendation?

Major revision

Comments to the Author(s)

Dear Catherine and colleagues,

This looks like a great project and I'd love to see it conducted. I've reviewed a couple of registered reports now and I always just want to know the results!! I'm not very patient ☐ Despite the significant amount of work that's been done in this area, a well-controlled and, particularly, well-powered studied is needed. So I think examining motion processing with suitable control conditions is a great line of enquiry.

I did have a bunch of thoughts when reading through. I'll mention a few of the major ones here which are backed up in specific comments below. But I'll make some comments on the required sections first.

Best wishes,
Nic Badcock

PS If you would like to clarify anything or even discuss these or other ideas related to the project, I'm very happy to be contacted (nicholas.badcock@uwa.edu.au) provided the editors are happy with this - feels to be in the spirit of collaborative Open Science.

Please comment explicitly on each of the following points in your comments to the authors:

1. The scientific validity of the research question(s)
The nature of the perceptual issues in dyslexia and how these relate to reading ability have been extremely evasive for researchers. A well-controlled and well-powered study such as the proposal will help to add clarity to the mixed literature. Therefore this is certainly scientific validity in the proposed questions.
2. The logic, rationale, and plausibility of the proposed hypotheses
The general logic is sound.
3. The soundness and feasibility of the methodology and analysis pipeline (including statistical power analysis where applicable)

I have some concerns about the nature of the developmental differences (e.g., executive control/function) with such a wide age range (8 to 14 years) but, perhaps even more critically, the nature of the reading difficulties. Heterogeneity is rampant within the dyslexic literature therefore I'd encourage the authors carefully consider the role of subtypes in the introduction and predictions - I've included some explanation of this in the further comments.

There are minor details of the methods that I think could be reviewed – for example, I'd like to see reaction times collected. More details below.

4. Whether the clarity and degree of methodological detail would be sufficient to replicate exactly the proposed experimental procedures and analysis pipeline

More information could be recorded about the precise timing of the displays – i.e., if the fixation on screen for a specified duration? What is the nature of the inter-trial interval? Does the display initiate with a keypress? It would be helpful to detailed all of these elements to ensure replication could be carried out.

The data handling and statistics are clearly defined.

5. Whether the authors provide a sufficiently clear and detailed description of the methods to prevent undisclosed flexibility in the experimental procedures or analysis pipeline

Yes. This looks fine. I did have a question (specified below) about the calculation of the thresholds from the staircase. And I would like the authors to consider non-parametric alternatives in the event of non-normally distributed data, rather than replacement.

6. Whether the authors have considered sufficient outcome-neutral conditions (e.g. positive controls) for ensuring that the results obtained are able to test the stated hypotheses

I think the control condition should do the trick. In addition, the authors might want to consider a general estimate of vigilance. Previous work has shown that controlling for this can account for differences in perceptual/attentional tasks (McLean et al., 2010)

Major comments/thoughts

1. Subtypes

There's a real need in this area and in dyslexic research more generally to be very carefully specifying the nature of the reading difficulty. I give more information on this below but I think it would be beneficial to the project to consider this carefully within the context of existing literature and ensure that the subtype or subtypes of reading difficulties that you're sampling will best answer the question.

2. Autism

It sounds really good to have a comparisons with another developmental disorder but the rationale for this inclusion wasn't clear to me. Felt like an opportunity for exploration rather than a theoretically-driven aspect of the work. Given that autism doesn't feature in the hypotheses or analysis, I think the value of this as part of the pre-registration needs to be re-evaluated for its centrality to the questions you're asking. It might be another story that can be answered in another paper.

3. Assessing reading in controls

I appreciate that some of the control data has already been collected but I feel that failing to collect reading and spelling data in controls is a flaw in the design. It's critical to demonstrate that the groups differ on these key abilities in order to draw the inferences we need from this work. In addition, I think it will be a missed opportunity if we aren't able to relate the perceptual performance back to reading/spelling abilities in a correlational/regression analysis. Including these measures and registering an analysis would add significant value to the project.

Specific comments:

4. Page 3, lines 5/6: Hyphenation – e.g., motion-coherence thresholds

> I've found that liberal use of hyphens for compound adjectives can be really helpful as a reader. Please consider adding a few throughout the manuscript.

5. Page 5, lines 43/44 "...we will be able to not only understand..."

> I think 'better understand' would be more accurate. This sounds like a solid study, but I'm not sure it'll explain everything!

6. Page 5, autism

> It'd be good to include more about why it's interesting to compare dyslexic and autistic observers – what's the theory here?

7. Page 7, Power analysis

> Useful to confirm which analysis this was based on?

8. Page 8, 8 to 14 years of age:

> This is a broad age-range. Please include a justification for this - there's a lot of general cognitive development going on across these ages. Will this be factored into the analysis? Was this included in the power analysis?

9. Page 8, composite score of 89 or less:

> This seems like quite a high cut off. I appreciate you've referenced Snowling's work for this but it would be useful to include the justification here. Typically I'd expect to be below one standard deviation which I assume would be 85 for a test like this.

10. Page 8, speeded phonemic decoding + spelling

> Extremely useful to include information/discussion of subtypes in the introduction. Some references to consider (Castles & Coltheart, 1993; Jones et al., 2011; Kohnen et al., 2012; McArthur et al., 2013). You might also want to include the sight word reading subscale of the TOWRE to help pinpoint sub skills (i.e., Coltheart et al., 2001; Ziegler et al., 2008). And, just to be sure that you know about it, there's a freely available non-speeded reading test of these subskills (Castles et al., 2009).

11. Page 7, measure reading and spelling in controls

> I think it's super important to know how your controls perform on these tasks. We really need to be able to establish the group differences but it would be useful to be able to relate these skills back to your perceptual tasks (i.e., with correlations or regressions).

12. Page 8, exclusions with replacement on perceptual tasks?

> Just wanted to check whether there's the intention to recruit more individuals with dyslexia (and controls) if they're excluded from the perceptual tasks.

13. Page 10, keyboard responses from children for reaction times?

> There would be benefit in having children press buttons/keys for their responses. It'll likely save time but also allow for reaction time to be included in the data set. This may be informative for other analyses of the data, but could even be factored in as a covariate. Given that the youngest children will be 8, this should be easily achieved.

14. Page 11, new reading/IQ data

> reading, as well as general capacity, can vary a lot across time. Given that the TOWRE-2 has 4 (or more?) parallel forms, I'd strongly encourage you to collect up-to-date data on this. To best examine the relationship between reading and perception, the closer together in time the measurements are taken the better.

15. Page 12, threshold estimates

> apologies if I missed this but I wasn't sure how the thresholds were calculated. This is typical based on the final X reversals of a staircase. It would be good to clearly define this (sticking with an even number of reversals to avoid bias)

16. Page 14, replacing extreme values

> One of the characteristics of perceptual performance in dyslexia is extreme values (Roach et al., 2004). I appreciate the up-front specification of these adjustments for pre-registration but I am concerned that we might be 'throwing the baby out with the bath water'. Please consider non-parametric alternatives for the analyses - perhaps just as a contingency plan if that data are not normally distributed.

17. Page 14, statistical analyses - what about autism?

> Although none of the hypotheses speak to autism, comparisons are mentioned in the introduction. Given that this is a registered report, I think it's be useful to specify how these comparisons will be treated.

> It also feels remiss to not make some predictions (or plan to explore) about the relationships between reading and perception. Ideally this would be conducted across the entire dataset (i.e., dyslexic and typical readers) but this hinges on collecting their reading data.

References

- Castles, A., & Coltheart, M. (1993). Varieties of developmental dyslexia. *Cognition*, 47(2), 149–180. [https://doi.org/10.1016/0010-0277\(93\)90003-E](https://doi.org/10.1016/0010-0277(93)90003-E)
- Castles, A., Coltheart, M., Larsen, L., Jones, P., Saunders, S., & McArthur, G. (2009). Assessing the basic components of reading: A revision of the Castles and Coltheart test with new norms. *Australian Journal of Learning Difficulties*, 14(1), 67–88. <https://doi.org/10.1080/19404150902783435>
- Coltheart, M., Rastle, K., Perry, C., Langdon, R., & Ziegler, J. (2001). DRC: A dual route cascaded model of visual word recognition and reading aloud. *Psychological Review*, 108(1), 204–256.
- Jones, K., Castles, A., & Kohnen, S. (2011). Subtypes of developmental reading disorders: Recent developments and directions for treatment. *Acquiring Knowledge in Speech, Language and Hearing*, 13(2), 79–83.
- Kohnen, S., Nickels, L., Castles, A., Friedmann, N., & McArthur, G. (2012). When 'slime' becomes 'smile': Developmental letter position dyslexia in English. *Neuropsychologia*, 50(14), 3681–3692. <https://doi.org/10.1016/j.neuropsychologia.2012.07.016>
- McArthur, G. M., Kohnen, S., Larsen, L., Jones, K., Anandakumar, T., Banales, E., & Castles, A. (2013). Getting to grips with the heterogeneity of developmental dyslexia. *Cognitive Neuropsychology*, 30(1), 1–24. <https://doi.org/10.1080/02643294.2013.784192>
- McLean, G. M. T., Castles, A., Coltheart, V., & Stuart, G. W. (2010). No evidence for a prolonged attentional blink in developmental dyslexia. *Cortex*, 46(10), 1317–1329. <https://doi.org/10.1016/j.cortex.2010.06.010>
- Roach, N. W., Edwards, V. T., & Hogben, J. H. (2004). The Tale is in the Tail: An Alternative Hypothesis for Psychophysical Performance Variability in Dyslexia. *Perception*, 33(7), 817–830. <https://doi.org/10.1068/p5207>
- Ziegler, J. C., Castel, C., Pech-Georgel, C., George, F., Alario, F.-X., & Perry, C. (2008). Developmental dyslexia and the dual route model of reading: Simulating individual differences and subtypes. *Cognition*, 107(1), 151–178. <https://doi.org/10.1016/j.cognition.2007.09.004>

Decision letter (RSOS-200414.R0)

Dear Dr Manning

On behalf of the Editors, I am pleased to inform you that your Manuscript RSOS-200414 entitled "Integration of visual motion and orientation signals in dyslexic children" deemed suitable for in-principle acceptance in Royal Society Open Science subject to minor revision in accordance with the referee and editor suggestions. Please find their comments at the end of this email.

The reviewers and handling editors have recommended publication, but also suggest some minor revisions to your manuscript. Therefore, I invite you to respond to the comments and revise your manuscript.

Full author guidelines can be found here <https://royalsocietypublishing.org/rsos/registered-reports#ReviewerGuideRegRep>.

Kind regards
Andrew Dunn
Royal Society Open Science
openscience@royalsociety.org

on behalf of Professor Chris Chambers
(Subject Editor, Royal Society Open Science)
openscience@royalsociety.org

Associate Editor Comments to Author (Professor Chris Chambers):

At the outset please accept my apologies for the slow handling your submission. This is in no way the fault of the reviewers who have supplied assessments (both of whom were on time), but rather due to another reviewer accepting the assignment but then becoming non-responsive. This caused a significant delay while we sought an alternative reviewer.

The good news is that the two expert reviewers who have assessed the manuscript find merit in the proposal, while also offering a range of constructive suggestions to consider in revision -- chiefly to clarify specific aspects of the design, consider additional measurements (and predictions of relevance), expansion of methodological detail, and addressing potential flaws (e.g. Reviewer 2, major point 3). In revising the proposal, please also include a Study Design template to show the clearest possible mapping between the hypotheses, sampling plans, analysis plans, and contingent interpretation given different outcomes. As a guide, I have attached a couple of examples of such tables from existing submissions approved at Royal Society Open Science.

Reviewer comments to Author:

Reviewer: 1

Comments to the Author(s)

In this report, Manning et. al. applied a research proposal using averaging and coherence tasks with motion direction or static orientation information to address the mechanisms of atypical global motion processing in dyslexic individuals. In my view, this will be an interesting study. Since the tasks provided here have already been applied in autistic individuals, I just have the following minor comments that I would like the authors to address:

1. For the including of dyslexic participants, are there any Guidelines on the treatment and diagnosis of dyslexia that author refers to? Please add.
2. During the experiment, is there any method to fix children's head and/or maintain their fixation? If not, will the uncertain of fixation or head movement add additional noise?

3. In statistical analysis, independent samples t-test would also be added to ensure the two groups are age-matched.

Reviewer: 2

Comments to the Author(s)

Dear Catherine and colleagues,

This looks like a great project and I'd love to see it conducted. I've reviewed a couple of registered reports now and I always just want to know the results!! I'm not very patient ☐ Despite the significant amount of work that's been done in this area, a well-controlled and, particularly, well-powered study is needed. So I think examining motion processing with suitable control conditions is a great line of enquiry.

I did have a bunch of thoughts when reading through. I'll mention a few of the major ones here which are backed up in specific comments below. But I'll make some comments on the required sections first.

Best wishes,

Nic Badcock

PS If you would like to clarify anything or even discuss these or other ideas related to the project, I'm very happy to be contacted (nicholas.badcock@uwa.edu.au) provided the editors are happy with this - feels to be in the spirit of collaborative Open Science.

Please comment explicitly on each of the following points in your comments to the authors:

1. The scientific validity of the research question(s)

The nature of the perceptual issues in dyslexia and how these relate to reading ability have been extremely evasive for researchers. A well-controlled and well-powered study such as the proposal will help to add clarity to the mixed literature. Therefore this is certainly scientific validity in the proposed questions.

2. The logic, rationale, and plausibility of the proposed hypotheses

The general logic is sound.

3. The soundness and feasibility of the methodology and analysis pipeline (including statistical power analysis where applicable)

I have some concerns about the nature of the developmental differences (e.g., executive control/function) with such a wide age range (8 to 14 years) but, perhaps even more critically, the nature of the reading difficulties. Heterogeneity is rampant within the dyslexic literature therefore I'd encourage the authors carefully consider the role of subtypes in the introduction and predictions - I've included some explanation of this in the further comments.

There are minor details of the methods that I think could be reviewed - for example, I'd like to see reaction times collected. More details below.

4. Whether the clarity and degree of methodological detail would be sufficient to replicate exactly the proposed experimental procedures and analysis pipeline

More information could be recorded about the precise timing of the displays - i.e., if the fixation on screen for a specified duration? What is the nature of the inter-trial interval? Does the display initiate with a keypress? It would be helpful to detailed all of these elements to ensure replication could be carried out.

The data handling and statistics are clearly defined.

5. Whether the authors provide a sufficiently clear and detailed description of the methods to prevent undisclosed flexibility in the experimental procedures or analysis pipeline

Yes. This looks fine. I did have a question (specified below) about the calculation of the thresholds from the staircase. And I would like the authors to consider non-parametric alternatives in the event of non-normally distributed data, rather than replacement.

6. Whether the authors have considered sufficient outcome-neutral conditions (e.g. positive controls) for ensuring that the results obtained are able to test the stated hypotheses

I think the control condition should do the trick. In addition, the authors might want to consider a general estimate of vigilance. Previous work has shown that controlling for this can account for differences in perceptual/attentional tasks (McLean et al., 2010)

Major comments/thoughts

1. Subtypes

There's a real need in this area and in dyslexic research more generally to be very carefully specifying the nature of the reading difficulty. I give more information on this below but I think it would be beneficial to the project to consider this carefully within the context of existing literature and ensure that the subtype or subtypes of reading difficulties that you're sampling will best answer the question.

2. Autism

It sounds really good to have a comparisons with another developmental disorder but the rationale for this inclusion wasn't clear to me. Felt like an opportunity for exploration rather than a theoretically-driven aspect of the work. Given that autism doesn't feature in the hypotheses or analysis, I think the value of this as part of the pre-registration needs to be re-evaluated for its centrality to the questions you're asking. It might be another story that can be answered in another paper.

3. Assessing reading in controls

I appreciate that some of the control data has already been collected but I feel that failing to collect reading and spelling data in controls is a flaw in the design. It's critical to demonstrate that the groups differ on these key abilities in order to draw the inferences we need from this work. In addition, I think it will be a missed opportunity if we aren't able to relate the perceptual performance back to reading/spelling abilities in a correlational/regression analysis. Including these measures and registering an analysis would add significant value to the project.

Specific comments:

4. Page 3, lines 5/6: Hyphenation – e.g., motion-coherence thresholds

> I've found that liberal use of hyphens for compound adjectives can be really helpful as a reader. Please consider adding a few throughout the manuscript.

5. Page 5, lines 43/44 " ...we will be able to not only understand..."

> I think 'better understand' would be more accurate. This sounds like a solid study, but I'm not sure it'll explain everything!

6. Page 5, autism

> It'd be good to include more about why it's interesting to compare dyslexic and autistic observers – what's the theory here?

7. Page 7, Power analysis

> Useful to confirm which analysis this was based on?

8. Page 8, 8 to 14 years of age:

> This is a broad age-range. Please include a justification for this - there's a lot of general cognitive development going on across these ages. Will this be factored into the analysis? Was this included in the power analysis?

9. Page 8, composite score of 89 or less:

> This seems like quite a high cut off. I appreciate you've referenced Snowling's work for this but it would be useful to include the justification here. Typically I'd expect to be below one standard deviation which I assume would be 85 for a test like this.

10. Page 8, speeded phonemic decoding + spelling

> Extremely useful to include information/discussion of subtypes in the introduction. Some references to consider (Castles & Coltheart, 1993; Jones et al., 2011; Kohnen et al., 2012; McArthur et al., 2013). You might also want to include the sight word reading subscale of the TOWRE to help pinpoint sub skills (i.e., Coltheart et al., 2001; Ziegler et al., 2008). And, just to be sure that you know about it, there's a freely available non-speeded reading test of these subskills (Castles et al., 2009).

11. Page 7, measure reading and spelling in controls

> I think it's super important to know how your controls perform on these tasks. We really need to be able to establish the group differences but it would be useful to be able to relate these skills back to your perceptual tasks (i.e., with correlations or regressions).

12. Page 8, exclusions with replacement on perceptual tasks?

> Just wanted to check whether there's the intention to recruit more individuals with dyslexia (and controls) if they're excluded from the perceptual tasks.

13. Page 10, keyboard responses from children for reaction times?

> There would be benefit in having children press buttons/keys for their responses. It'll likely save time but also allow for reaction time to be included in the data set. This may be informative for other analyses of the data, but could even be factored in as a covariate. Given that the youngest children will be 8, this should be easily achieved.

14. Page 11, new reading/IQ data

> reading, as well as general capacity, can vary a lot across time. Given that the TOWRE-2 has 4 (or more?) parallel forms, I'd strongly encourage you to collect up-to-date data on this. To best examine the relationship between reading and perception, the closer together in time the measurements are taken the better.

15. Page 12, threshold estimates

> apologies if I missed this but I wasn't sure how the thresholds were calculated. This is typical based on the final X reversals of a staircase. It would be good to clearly define this (sticking with an even number of reversals to avoid bias)

16. Page 14, replacing extreme values

> One of the characteristics of perceptual performance in dyslexia is extreme values (Roach et al., 2004). I appreciate the up-front specification of these adjustments for pre-registration but I am concerned that we might be 'throwing the baby out with the bath water'. Please consider non-parametric alternatives for the analyses - perhaps just as a contingency plan if that data are not normally distributed.

17. Page 14, statistical analyses - what about autism?

> Although none of the hypotheses speak to autism, comparisons are mentioned in the introduction. Given that this is a registered report, I think it's be useful to specify how these comparisons will be treated.

> It also feels remiss to not make some predictions (or plan to explore) about the relationships between reading and perception. Ideally this would be conducted across the entire dataset (i.e., dyslexic and typical readers) but this hinges on collecting their reading data.

References

- Castles, A., & Coltheart, M. (1993). Varieties of developmental dyslexia. *Cognition*, 47(2), 149-180. [https://doi.org/10.1016/0010-0277\(93\)90003-E](https://doi.org/10.1016/0010-0277(93)90003-E)
- Castles, A., Coltheart, M., Larsen, L., Jones, P., Saunders, S., & McArthur, G. (2009). Assessing the basic components of reading: A revision of the Castles and Coltheart test with new norms. *Australian Journal of Learning Difficulties*, 14(1), 67-88. <https://doi.org/10.1080/19404150902783435>
- Coltheart, M., Rastle, K., Perry, C., Langdon, R., & Ziegler, J. (2001). DRC: A dual route cascaded model of visual word recognition and reading aloud. *Psychological Review*, 108(1), 204-256.
- Jones, K., Castles, A., & Kohnen, S. (2011). Subtypes of developmental reading disorders: Recent developments and directions for treatment. *Acquiring Knowledge in Speech, Language and Hearing*, 13(2), 79-83.

- Kohnen, S., Nickels, L., Castles, A., Friedmann, N., & McArthur, G. (2012). When 'slime' becomes 'smile': Developmental letter position dyslexia in English. *Neuropsychologia*, 50(14), 3681–3692. <https://doi.org/10.1016/j.neuropsychologia.2012.07.016>
- McArthur, G. M., Kohnen, S., Larsen, L., Jones, K., Anandakumar, T., Banales, E., & Castles, A. (2013). Getting to grips with the heterogeneity of developmental dyslexia. *Cognitive Neuropsychology*, 30(1), 1–24. <https://doi.org/10.1080/02643294.2013.784192>
- McLean, G. M. T., Castles, A., Coltheart, V., & Stuart, G. W. (2010). No evidence for a prolonged attentional blink in developmental dyslexia. *Cortex*, 46(10), 1317–1329. <https://doi.org/10.1016/j.cortex.2010.06.010>
- Roach, N. W., Edwards, V. T., & Hogben, J. H. (2004). The Tale is in the Tail: An Alternative Hypothesis for Psychophysical Performance Variability in Dyslexia. *Perception*, 33(7), 817–830. <https://doi.org/10.1068/p5207>
- Ziegler, J. C., Castel, C., Pech-Georgel, C., George, F., Alario, F.-X., & Perry, C. (2008). Developmental dyslexia and the dual route model of reading: Simulating individual differences and subtypes. *Cognition*, 107(1), 151–178. <https://doi.org/10.1016/j.cognition.2007.09.004>

Author's Response to Decision Letter for (RSOS-200414.R0)

See Appendix A.

RSOS-200414.R1 (Revision)

Review form: Reviewer 1 (Jiawei Zhou)

Do you have any ethical concerns with this paper?

No

Recommendation?

Accept in principle

Comments to the Author(s)

The authors addressed the concerns that I raised, thank you.

Review form: Reviewer 2 (Nic Badcock)

Do you have any ethical concerns with this paper?

No

Recommendation?

Accept in principle

Comments to the Author(s)

Dear Cathy and colleagues,

Thank you and well done on the explanations of the approach/rationale and changes to the manuscript. I now have a much better appreciation for where you're coming from on a number of the point and this is reflected in the updated proposal. You're arguments are nice and defensible and you've clarified that a number of my comments related to different questions (i.e individual differences) that aren't the focus of the work. A few very minor things below.

Best of luck with it!

Nic Badcock

Note: page numbers are those in the header of the document (i.e. not PDF pages)

1. Page 3, Line 14/16: The Elliot & Grigorenko citation is missing its year
2. Page 4, Lines 23/29: I appreciate the mention of subtypes. I think more could be done to contextualise this within the current research. This is a minor point but (a) it sticks out as a little in this paragraph: consider re-locating or leading it with something like "A further consideration for inconsistencies in this literature..." and (b) potentially one further/final sentence saying more or less what was in the response letter (still need to establish the overall effect before we can dig deeper) would round this out so that the reader understood the rationale for this comment.
3. Page 5, Line 54: I think 'better understand' rather than 'understand better' reads more naturally

Decision letter (RSOS-200414.R1)

Dear Dr Manning

On behalf of the Editor, I am pleased to inform you that your Manuscript RSOS-200414.R1 entitled "Integration of visual motion and orientation signals in dyslexic children" has been accepted in principle for publication in Royal Society Open Science. The reviewers' and editors' comments are included at the end of this email.

You may now progress to Stage 2 and complete the study as approved. Before commencing data collection we ask that you:

- 1) Update the journal office as to the anticipated completion date of your study.
- 2) Register your approved protocol on the Open Science Framework (<https://osf.io/rr>) or other recognised repository, either publicly or privately under embargo until submission of the Stage 2 manuscript. Please note that a time-stamped, independent registration of the protocol is mandatory under journal policy, and manuscripts that do not conform to this requirement cannot be considered at Stage 2. The protocol should be registered unchanged from its current approved state, with the time-stamp preceding implementation of the approved study design. We strongly recommend using the dedicated RR registration portal supported by the OSF at <https://osf.io/rr>

Following completion of your study, we invite you to resubmit your paper for peer review as a Stage 2 Registered Report. Please note that your manuscript can still be rejected for publication at Stage 2 if the Editors consider any of the following conditions to be met:

- The results were unable to test the authors' proposed hypotheses by failing to meet the approved outcome-neutral criteria.

- The authors altered the Introduction, rationale, or hypotheses, as approved in the Stage 1 submission.
- The authors failed to adhere closely to the registered experimental procedures. Please note that any deviations from the approved experimental procedures must be communicated to the editor immediately for approval, and prior to the completion of data collection. Failure to do so can result in revocation of in-principle acceptance and rejection at Stage 2 (see complete guidelines for further information).
- Any post-hoc (unregistered) analyses were either unjustified, insufficiently caveated, or overly dominant in shaping the authors' conclusions.
- The authors' conclusions were not justified given the data obtained.

We encourage you to read the complete guidelines for authors concerning Stage 2 submissions at <https://royalsocietypublishing.org/rsos/registered-reports#ReviewerGuideRegRep>. Please especially note the requirements for data sharing, reporting the URL of the independently registered protocol, and that withdrawing your manuscript will result in publication of a Withdrawn Registration.

Please note that Royal Society Open Science will introduce article processing charges for all new submissions received from 1 January 2018. Registered Reports submitted and accepted after this date will ONLY be subject to a charge if they subsequently progress to and are accepted as Stage 2 Registered Reports. If your manuscript is submitted and accepted for publication after 1 January 2018 (i.e. as a full Stage 2 Registered Report), you will be asked to pay the article processing charge, unless you request a waiver and this is approved by Royal Society Publishing. You can find out more about the charges at <https://royalsocietypublishing.org/rsos/charges>. Should you have any queries, please contact openscience@royalsociety.org.

Once again, thank you for submitting your manuscript to Royal Society Open Science and we look forward to receiving your Stage 2 submission. If you have any questions at all, please do not hesitate to get in touch. We look forward to hearing from you shortly with the anticipated submission date for your stage two manuscript.

on behalf of Chris Chambers (Registered Reports Editor, Royal Society Open Science)
openscience@royalsociety.org

Associate Editor Comments to Author (Professor Chris Chambers):

Associate Editor: 1

Comments to the Author:

Both reviewers are now satisfied with the Stage 1 submission and recommend in-principle acceptance. Reviewer 2 offers some helpful suggestions for minor amendments to improve clarity. I am happy for the authors to make these small changes and then register the protocol formally at <https://osf.io/rr/> as instructed above, without submitting a revised Stage 1 manuscript.

Reviewers' comments to Author:

Reviewer: 1

Comments to the Author(s)

The authors addressed the concerns that I raised, thank you.

Reviewer: 2

Comments to the Author(s)

Dear Cathy and colleagues,

Thank you and well done on the explanations of the approach/rationale and changes to the manuscript. I now have a much better appreciation for where you're coming from on a number of the point and this is reflected in the updated proposal. You're arguments are nice and defensible and you've clarified that a number of my comments related to different questions (i.e individual differences) that aren't the focus of the work. A few very minor things below.

Best of luck with it!

Nic Badcock

Note: page numbers are those in the header of the document (i.e. not PDF pages)

1. Page 3, Line 14/16: The Elliot & Grigorenko citation is missing its year
2. Page 4, Lines 23/29: I appreciate the mention of subtypes. I think more could be done to contextualise this within the current research. This is a minor point but (a) it sticks out as a little in this paragraph: consider re-locating or leading it with something like "A further consideration for inconsistencies in this literature..." and (b) potentially one further/final sentence saying more or less what was in the response letter (still need to establish the overall effect before we can dig deeper) would round this out so that the reader understood the rationale for this comment.
3. Page 5, Line 54: I think 'better understand' rather than 'understand better' reads more naturally

Author's Response to Decision Letter for (RSOS-200414.R1)

See Appendix B.

RSOS-200414.R2 (Revision)

Review form: Reviewer 2 (Nic Badcock)

Is the manuscript scientifically sound in its present form?

Yes

Are the interpretations and conclusions justified by the results?

Yes

Is the language acceptable?

Yes

Do you have any ethical concerns with this paper?

No

Have you any concerns about statistical analyses in this paper?

No

Recommendation?

Accept with minor revision

Comments to the Author(s)

Dear Cathy and colleagues,

Firstly, well done for completing this project during COVID times. It's impressive in general in terms of your timeframe, but even more so under the circumstances. As mentioned below, I was surprised by the small magnitude of the effects, but that's one of the benefits of preregistration. The updates are nicely presented and clear. I have a few comments for consideration below – always happy to clarify/discuss over email if that's helpful.

Best wishes,

Nic Badcock

Required comments:

1. Whether the data are able to test the authors' proposed hypotheses by passing the approved outcome-neutral criteria (such as absence of floor and ceiling effects or success of positive controls)

> I confirm that the data are suitable to test the hypotheses and there is sufficient variability in the data

2. Whether the Introduction, rationale and stated hypotheses are the same as the approved Stage 1 submission

> This is consistent – modifications for past tense have been made as appropriate

3. Whether the authors adhered precisely to the registered experimental procedures

> I confirm this to be the case

4. Where applicable, whether any unregistered exploratory statistical analyses are justified, methodologically sound, and informative

> The additional analyses are a useful inclusion, clarifying some pertinent queries related to the sampling with respect to previous work

5. Whether the authors' conclusions are justified given the data

> The conclusions fit nicely. As I'll note below, it would be useful to temper the absolute nature of significance with the subtleties of effect sizes and inconclusive/weak Bayesian evidence.

General comments:

6. Magnitude of effect

One of the great benefits to the preregistration is that we get to see that magnitude of effects in a transparent investigation. I was surprised by the small effect sizes reported, especially within the context of the Cohen's d value used for the power analysis. This surprise is in no way a criticism – I imagine it reflective of the true effects in a heterogenous sample, as you note in the Discussion. I would like to see the magnitude of the effects highlighted in the Abstract and Discussion to a higher degree. I feel/fear that the black and white nature of NHST is too extreme in this case, particularly in the Abstract, and including a comment on the nature of the effect size of the significant effects, and Bayesian evidence if it fits within the word count, would do better service to the data.

To be sure, this magnitude is reflected in the Discussion (e.g., line 532: "...to note that the group differences were subtle..." and 574: "...the relatively inconclusive nature of our results...") but it may be worth having an explicit sentence or two in the opening paragraph dedicated to confirming the small magnitude of effects in relation to the basis for the power analysis. It could be flagged as a limitation if you see fit – again, not so much a criticism of the study, but the literature upon which it was based.

7. COVID considerations

This is a very light comment, but I wondered whether there's any sense in the literature regarding sampling biases with respect to who's willing to participate in research during a pandemic. My impression is that the UK, certainly more so than Australia, has been business as usual, but it seems prudent to suggest a little check of the literature with an eye for a potential bias of whether individuals with more extreme difficulties (in reading, in general) may be avoiding testing during these times. As mentioned, a very light comment, I'm not aware of anything, but it could be worth a quick look. I note that the reading scores (Table 1) are as you might expect, and it's hard to imagine that poor readers with particularly problematic magno

deficits would somehow avoid testing – so perhaps it's just worth a reminder in the limitations that the testing was conducted during COVID times which may have influenced the population willing to participate. Only if this seems at all relevant in your opinion.

Specific comments:

8. Page 6, line 147: "2. We investigated whether dyslexic children show atypical sampling in the motion- ..." > Please consider rephrasing this to be in line with the other hypotheses (i.e., We hypothesised that ...) – as included in Table 2.

9. Page 7, line 167: same comment as previous regarding item number 6. Apologies if I've missed something here. I guess either way, some clarification in the wording would help slow people like me (I also appreciate that these are comments related to the previously accepted portion of the manuscript – hopefully it's okay to comment on this)

10. Page 22/23, Line 397 – 399: Table 1 header row > Consider adding an extra row for the column headers with M (SD) Error Range for the Typically Developing and Dyslexic columns – so twice. It's a bit repetitive but I feel it'll make the table more accessible. The note under the table feels less intuitive than additional column headers to me. Additionally, consider confirming in the table title that the error range reflects a portion – my first impression is that error range is often a count.

11. Page 24, Figure 3 > Great to see all the data points on these plots! Please consider capitalising the group names on the x axes, just for presentation purposes, feels ever so slightly under baked with lower case (though I always prefer to code in lower case, so I appreciate where it comes from).

12. Page 28, lines 512/513: "This analysis revealed inconclusive evidence for or against the remaining hypothesised group differences." > my reading is that 'the remaining' might be better replaced by 'all' in this instance. Please consider whether this might fit.

13. Page 32, line 621: > Could you comment on how your age distribution compares to other research where bigger effects were found?

Decision letter (RSOS-200414.R2)

Dear Dr Manning:

On behalf of the Editor, I am pleased to inform you that your Stage 2 Registered Report RSOS-200414.R2 entitled "Integration of visual motion and orientation signals in dyslexic children: An equivalent noise approach" has been deemed suitable for publication in Royal Society Open Science subject to minor revision in accordance with the referee suggestions. Please find the referees' comments at the end of this email.

The reviewers and Subject Editor have recommended publication, but also suggest some minor revisions to your manuscript. We invite you to respond to the comments and revise your manuscript. Below the referees' and Editors' comments (where applicable) we provide additional requirements. Final acceptance of your manuscript is dependent on these requirements being met. We provide guidance below to help you prepare your revision.

Please submit your revised manuscript and required files (see below) no later than 7 days from today's (ie 16-Mar-2022) date. Note: the ScholarOne system will 'lock' if submission of the revision is attempted 7 or more days after the deadline. If you do not think you will be able to meet this deadline please contact the editorial office immediately.

on behalf of Professor Chris Chambers
(Registered Reports Editor, Royal Society Open Science)
openscience@royalsociety.org

Associate Editor Comments to Author (Professor Chris Chambers):

Associate Editor: 1

Comments to the Author:

One of the original reviewers from Stage 1 kindly returned to evaluate the Stage 2 manuscript, and I have decided that we can continue with an interim decision based on this reviewer's assessment and my own reading of the manuscript. As will see, the reviewer is broadly very positive about your submission and notes some minor areas that would benefit from elaboration or at least further consideration in a response. I concur and agree that this is a well conducted and reported RR, under challenging circumstances.

I will assess the revised manuscript and response at desk, and provided you are able to respond to all points thoroughly then I anticipate that full acceptance should be forthcoming without requiring further in-depth review.

Comments to Author:

Reviewer: 2

Comments to the Author(s)

Dear Cathy and colleagues,

Firstly, well done for completing this project during COVID times. It's impressive in general in terms of your timeframe, but even more so under the circumstances. As mentioned below, I was surprised by the small magnitude of the effects, but that's one of the benefits of preregistration. The updates are nicely presented and clear. I have a few comments for consideration below – always happy to clarify/discuss over email if that's helpful.

Best wishes,

Nic Badcock

Required comments:

1. Whether the data are able to test the authors' proposed hypotheses by passing the approved outcome-neutral criteria (such as absence of floor and ceiling effects or success of positive controls)

> I confirm that the data are suitable to test the hypotheses and there is sufficient variability in the data

2. Whether the Introduction, rationale and stated hypotheses are the same as the approved Stage 1 submission

> This is consistent – modifications for past tense have been made as appropriate

3. Whether the authors adhered precisely to the registered experimental procedures

> I confirm this to be the case

4. Where applicable, whether any unregistered exploratory statistical analyses are justified, methodologically sound, and informative

> The additional analyses are a useful inclusion, clarifying some pertinent queries related to the sampling with respect to previous work

5. Whether the authors' conclusions are justified given the data

> The conclusions fit nicely. As I'll note below, it would be useful to temper the absolute nature of significance with the subtleties of effect sizes and inconclusive/weak Bayesian evidence.

General comments:

6. Magnitude of effect

One of the great benefits to the preregistration is that we get to see that magnitude of effects in a transparent investigation. I was surprised by the small effect sizes reported, especially within the context of the Cohen's d value used for the power analysis. This surprise is in no way a criticism – I imagine it reflective of the true effects in a heterogeneous sample, as you note in the Discussion. I would like to see the magnitude of the effects highlighted in the Abstract and Discussion to a higher degree. I feel/fear that the black and white nature of NHST is too extreme in this case, particularly in the Abstract, and including a comment on the nature of the effect size of the significant effects, and Bayesian evidence if it fits within the word count, would do better service to the data.

To be sure, this magnitude is reflected in the Discussion (e.g., line 532: "...to note that the group differences were subtle..." and 574: "...the relatively inconclusive nature of our results...") but it may be worth having an explicit sentence or two in the opening paragraph dedicated to confirming the small magnitude of effects in relation to the basis for the power analysis. It could be flagged as a limitation if you see fit – again, not so much a criticism of the study, but the literature upon which it was based.

7. COVID considerations

This is a very light comment, but I wondered whether there's any sense in the literature regarding sampling biases with respect to who's willing to participate in research during a pandemic. My impression is that the UK, certainly more so than Australia, has been business as usual, but it seems prudent to suggest a little check of the literature with an eye for a potential bias of whether individuals with more extreme difficulties (in reading, in general) may be avoiding testing during these times. As mentioned, a very light comment, I'm not aware of anything, but it could be worth a quick look. I note that the reading scores (Table 1) are as you might expect, and it's hard to imagine that poor readers with particularly problematic magnitudinal deficits would somehow avoid testing – so perhaps it's just worth a reminder in the limitations that the testing was conducted during COVID times which may have influenced the population willing to participate. Only if this seems at all relevant in your opinion.

Specific comments:

8. Page 6, line 147: "2. We investigated whether dyslexic children show atypical sampling in the motion- ..." > Please consider rephrasing this to be in line with the other hypotheses (i.e., We hypothesised that ...) – as included in Table 2.

9. Page 7, line 167: same comment as previous regarding item number 6. Apologies if I've missed something here. I guess either way, some clarification in the wording would help slow people like me 🤔🤔 (I also appreciate that these are comments related to the previously accepted portion of the manuscript – hopefully it's okay to comment on this)

10. Page 22/23, Line 397 – 399: Table 1 header row > Consider adding an extra row for the column headers with M (SD) Error Range for the Typically Developing and Dyslexic columns –

so twice. It's a bit repetitive but I feel it'll make the table more accessible. The note under the table feels less intuitive than additional column headers to me. Additionally, consider confirming in the table title that the error range reflects a portion – my first impression is that error range is often a count.

11. Page 24, Figure 3 > Great to see all the data points on these plots! Please consider capitalising the group names on the x axes, just for presentation purposes, feels ever so slightly under baked with lower case (though I always prefer to code in lower case, so I appreciate where it comes from).

12. Page 28, lines 512/513: "This analysis revealed inconclusive evidence for or against the remaining hypothesised group differences." > my reading is that 'the remaining' might be better replaced by 'all' in this instance. Please consider whether this might fit.

13. Page 32, line 621: > Could you comment on how your age distribution compares to other research where bigger effects were found?

===PREPARING YOUR MANUSCRIPT===

one version should clearly identify all the changes that have been made (for instance, in coloured highlight, in bold text, or tracked changes);

===PREPARING YOUR REVISION IN SCHOLARONE===

-- If you are requesting an article processing charge waiver, you must select the relevant waiver option (if requesting a discretionary waiver, the form should have been uploaded, see 'File upload' above).

-- If you have uploaded any electronic supplementary (ESM) files, please ensure you follow the guidance at <https://royalsociety.org/journals/authors/author-guidelines/#supplementary-material> to include a suitable title and informative caption. An example of appropriate titling and captioning may be found at https://figshare.com/articles/Table_S2_from_Is_there_a_trade-off_between_peak_performance_and_performance_breadth_across_temperatures_for_aerobic_scope_in_teleost_fishes_/3843624.

Author's Response to Decision Letter for (RSOS-200414.R2)

See Appendix C.

Decision letter (RSOS-200414.R3)

Dear Dr Manning:

It is a pleasure to accept your Stage 2 Registered Report entitled "Integration of visual motion and orientation signals in dyslexic children: An equivalent noise approach" in its current form for publication in Royal Society Open Science.

Thank you for your fine contribution. On behalf of the Editors of Royal Society Open Science, we look forward to your continued contributions to the journal.

on behalf of Professor Chris Chambers (Subject Editor)
openscience@royalsociety.org

Appendix A

Dr Catherine Manning
Department of Experimental Psychology
University of Oxford

Tel +44 (0)1865 271442
Email catherine.manning@psy.ox.ac.uk

30th June 2020

Professor Chris Chambers
Subject Editor, *Royal Society Open Science*

Dear Professor Chambers,

Please find enclosed a revised Stage 1 Registered Report titled *Integration of visual motion and orientation signals in dyslexic children* (RSOS-200414).

We were pleased to hear that our manuscript was deemed suitable for in-principle acceptance at *Royal Society Open Science* subject to minor revision. We thank you and the reviewers very much for the constructive comments. We outline our response to these comments, point-by-point, below, and have made appropriate adjustments throughout the manuscript (marked in red).

We hope that you will find this version suitable for In-Principle Acceptance at Royal Society Open Science.

We look forward to hearing from you again.

Yours sincerely,

Catherine Manning (on behalf of the authors)

EDITOR COMMENTS:

The good news is that the two expert reviewers who have assessed the manuscript find merit in the proposal, while also offering a range of constructive suggestions to consider in revision -- chiefly to clarify specific aspects of the design, consider additional measurements (and predictions of relevance), expansion of methodological detail, and addressing potential flaws (e.g. Reviewer 2, major point 3). In revising the proposal, please also include a Study Design template to show the clearest possible mapping between the hypotheses, sampling plans, analysis plans, and contingent interpretation given different outcomes. As a guide, I have attached a couple of examples of such tables from existing submissions approved at Royal Society Open Science.

RESPONSE: We thank the Editor for handling this submission. We outline our response to reviewers' comments below and have now included a Study Design Template (pp. 16-21).

REVIEWER 1 COMMENTS:

In this report, Manning et. al. applied a research proposal using averaging and coherence tasks with motion direction or static orientation information to address

the mechanisms of atypical global motion processing in dyslexic individuals. In my view, this will be an interesting study.

RESPONSE: Thank you for your encouraging comments.

1. For the including of dyslexic participants, are there any Guidelines on the treatment and diagnosis of dyslexia that author refers to? Please add.

RESPONSE: The definition of dyslexia is continually evolving with a lack of consensus (see Protopapas, 2019), but we use a definition in line with the British Dyslexia Association and Rose (2009). We have now added these references in the opening sentence (p. 3).

2. During the experiment, is there any method to fix children's head and/or maintain their fixation? If not, will the uncertain of fixation or head movement add additional noise?

RESPONSE: We thank the Reviewer for this question. Efforts to fix children's head position could make the session less tolerable for many children and lead to them terminating the session early. However, we now state that we will use a chin-rest (p. 12) to keep viewing distance consistent, as we did in our previous studies using this paradigm. In our first studies in typical development (Manning et al., 2014) and autism (Manning et al., 2015) we collected eyetracking data to measure fixation stability from a subset of participants. In our typically developing study, we found relationships between fixation stability and internal noise. However, it was only the youngest children (aged 5 years), who had less stable fixations than the adults, while the fixation stability of older children (7 years and up) were not significantly different from adults. In the autism study, we found no significant correlations between fixation stability and performance in either the autistic or typically developing children (aged 6 to 14 years). We therefore think that is unlikely that group differences in performance in our children will be substantially affected by differences in fixation stability in the proposed study. As in our previous studies, the experimenter will monitor children's fixation carefully throughout and provide regular reminders, only initiating trials when participants are attending. We have now stated this in the report (p. 12).

3. In statistical analysis, independent samples t-test would also be added to ensure the two groups are age-matched.

RESPONSE: Although we have used independent samples t-tests to assess age-matching in previous studies, we are now aware of the problems with using inferential statistics to assess group-matching. Such tests are not sufficient for establishing matching (Kover & Atwood, 2013) and should be used to draw inferences about a population rather than a sample (see Sassenhagen & Alday, 2016 and <https://janhove.github.io/reporting/2014/09/26/balance-tests>). Instead, we will 1) aim to match the ages of dyslexic and typically developing children as closely as possible during recruitment (now stated on p. 9), and 2) report descriptive statistics relating to age for each group, including differences in variances and standardised means (effect sizes). We may also look at age effects in exploratory analyses (note we have not mentioned this in the current report in line with the Registered Report recommendations which suggest that this blurs the distinction between confirmatory and exploratory hypotheses).

REVIEWER 2 COMMENTS:

This looks like a great project and I'd love to see it conducted. I've reviewed a couple of registered reports now and I always just want to know the results!! I'm not very patient. Despite the significant amount of work that's been done in this area, a well-controlled and, particularly, well-powered studied is needed. So I think examining motion processing with suitable control conditions is a great line of enquiry. I did have a bunch of thoughts when reading through. I'll mention a few of the major

ones here which are backed up in specific comments below. But I'll make some comments on the required sections first.

1. The scientific validity of the research question(s)

The nature of the perceptual issues in dyslexia and how these relate to reading ability have been extremely evasive for researchers. A well-controlled and well-powered study such as the proposal will help to add clarity to the mixed literature. Therefore this is certainly scientific validity in the proposed questions.

2. The logic, rationale, and plausibility of the proposed hypotheses

The general logic is sound.

RESPONSE: Thank you, Nic, for your enthusiasm and constructive comments.

3. The soundness and feasibility of the methodology and analysis pipeline (including statistical power analysis where applicable)

I have some concerns about the nature of the developmental differences (e.g., executive control/function) with such a wide age range (8 to 14 years) but, perhaps even more critically, the nature of the reading difficulties. Heterogeneity is rampant within the dyslexic literature therefore I'd encourage the authors carefully consider the role of subtypes in the introduction and predictions – I've included some explanation of this in the further comments.

RESPONSE: Having a relatively wide age range will allow us to reach the sample size required for this well-powered study – narrowing the age range would make this considerably more difficult. We now explain this on p. 7. Following our studies of typical development, we do expect some age-related differences within this age range (see Manning et al., 2014, Figure 3). However, we do not have any hypotheses regarding different rates of development in dyslexia and typical development. This study is the first study to apply the equivalent noise paradigm to a dyslexic population, and will help to determine the presence and/or absence of group differences. Following this, future studies will be able to investigate whether dyslexic and typically developing children follow different developmental trajectories. We can look at possible age-related effects in our sample in exploratory analyses, but we may be underpowered to determine how age relationships differ between the groups, given that there is considerable variability even within an age group (see again Manning et al., 2014, Figure 3).

For a similar reason, we have decided not to break the dyslexia group into subtypes. The first reason is practical, as subgroup analysis would have reduced power to detect differences. The second reason is theoretical, given that there is no strong evidence for the existence of subtypes with clearly distinct cognitive or biological profiles (Elliott & Grigorenko, 2014). Previous studies of motion perception also tend not to break the dyslexia sample into subgroups (n.b. one study that did break down into subgroups found elevated motion coherence thresholds across all subgroups; Ridder et al., 2001). As with the study of age effects, these are interesting questions that could be addressed in follow-up research, but we think it is important to first establish the presence or absence of effects in this paradigm with as large a sample of dyslexic children as possible. However, we now mention subtypes in the Introduction to show that we are aware of this debate (p. 4): "While distinct subtypes of dyslexia have been proposed (Castles & Coltheart, 1993; Jones et al., 2011; McArthur et al., 2013), there is currently no consensus that motion-coherence processing is differentially affected in these subtypes (see Ridder et al., 2001; Boden & Giaschi, 2007)". We intend to revisit these issues in the Discussion section.

There are minor details of the methods that I think could be reviewed – for example, I'd like to see reaction times collected. More details below.

RESPONSE: We see the advantages of collecting response time data in allowing additional analyses of the data, but we explain below why we think the experimenter should relay responses to the computer, as in our previous autism study.

**4. Whether the clarity and degree of methodological detail would be sufficient to replicate exactly the proposed experimental procedures and analysis pipeline
More information could be recorded about the precise timing of the displays – i.e., if the fixation on screen for a specified duration? What is the nature of the inter-trial interval? Does the display initiate with a keypress? It would be helpful to detailed all of these elements to ensure replication could be carried out.**

The data handling and statistics are clearly defined.

RESPONSE: Thank you for pointing out this omission. The trial is initiated with a keypress from the experimenter (to ensure that children are attending before the trial starts), and the fixation point is on the screen at all times, so that there is no fixed inter-trial interval (see changes on pp. 9-12).

5. Whether the authors provide a sufficiently clear and detailed description of the methods to prevent undisclosed flexibility in the experimental procedures or analysis pipeline

Yes. This looks fine. I did have a question (specified below) about the calculation of the thresholds from the staircase.

RESPONSE: QUEST returns the most probable Bayesian estimate of the threshold at the end of the procedure using the mean or mode of the posterior probability density function (Watson & Pelli, 1983). We use the mean of the posterior probability density function, following previous studies using the efficient version of the equivalent noise model (e.g., Tibber et al., 2014, 2015; Manning et al., 2015 etc). We now specify this in the methods section (p. 13). We have also added details on the starting parameters of the QUEST staircase (p. 11).

And I would like the authors to consider non-parametric alternatives in the event of non-normally distributed data, rather than replacement.

RESPONSE: Instead of replacement, we now state that we will use non-parametric equivalents in the event that the data are not normally distributed (see below and on p. 15).

6. Whether the authors have considered sufficient outcome-neutral conditions (e.g. positive controls) for ensuring that the results obtained are able to test the stated hypotheses

I think the control condition should do the trick. In addition, the authors might want to consider a general estimate of vigilance. Previous work has shown that controlling for this can account for differences in perceptual/attentional tasks (McLean et al., 2010)

RESPONSE: Thank you for this suggestion which we have considered carefully. We have read the McLean et al. (2010) paper including details of the continuous performance test to measure sustained attention. We have decided not to include this measure for three reasons: 1) the testing battery already includes what we consider to be the duration limit of computerised tasks for children, and it looks like this measure would take an additional 5 minutes, 2) we would not have this measure on previously tested typically developing children, limiting the conclusions we can make about relationships with sustained attention from this measure, and 3) we already have a measure of attentiveness within the task itself, in the form of catch trials, which will allow us to assess attentiveness during task performance across all children.

1. Subtypes

There's a real need in this area and in dyslexic research more generally to be very carefully specifying the nature of the reading difficulty. I give more information on this below but I think it would be beneficial to the project to consider this carefully

within the context of existing literature and ensure that the subtype or subtypes of reading difficulties that you're sampling will best answer the question.

RESPONSE: Thank you for encouraging us to think about this and for discussing the issue further over email. We agree that studying subtypes of dyslexia in relation to visual motion perception would be very interesting. However, most studies of coherent motion perception do not differentiate between different subtypes, and Ridder et al. (2001) reported that coherent motion difficulties were shown across subtypes. We therefore do not have any clear differential hypotheses for different subtypes. Moreover, breaking down our sample into subgroups would reduce power to detect group differences. We therefore think that it is important for this initial study using the equivalent noise paradigm to assess group differences without subtyping, to provide a basis for future studies. We now refer to subtypes in the Introduction (p. 4), and plan to revisit this issue in the Discussion. Scores on the phonological decoding and sight word reading and spelling subtests will be available so that future researchers will be able to investigate this further.

2. Autism

It sounds really good to have a comparisons with another developmental disorder but the rationale for this inclusion wasn't clear to me. Felt like an opportunity for exploration rather than a theoretically-driven aspect of the work. Given that autism doesn't feature in the hypotheses or analysis, I think the value of this as part of the pre-registration needs to be re-evaluated for its centrality to the questions you're asking. It might be another story that can be answered in another paper.

RESPONSE: We are increasingly interested in cross-syndrome comparisons, as without these it is impossible to know whether altered motion processing tells us anything specific about dyslexia (or any other condition) or whether it is a more general marker of atypical development - in line with the dorsal stream vulnerability theory (a point we make on p. 4). The focus on autism specifically is because this is the only other atypically developing child population that has been assessed with the paradigm we propose. We obtained unexpected results in autistic children, but we do not yet know whether these results are specific to autism. This issue was therefore a driving factor behind the motivation of this work (along with understanding motion perception difficulties in dyslexia in its own right). We think it is important to explain this in the Introduction, as we will undoubtedly re-visit this point in the Discussion section.

Given that we do not yet know the pattern of performance that dyslexic children will show compared to typically developing children, it is difficult to specify follow-up statistical analyses to directly compare the developmental conditions. Instead, we envisage making comparisons between the two studies in the Discussion section (e.g., autistic children show enhanced motion integration compared to typically developing children, whereas dyslexic children do not), for example by comparing effect sizes and mean differences. We have now clarified this on p. 5 ("to compare patterns of performance in dyslexic and autistic children relative to typically developing children") and a further justification: "If the pattern found in dyslexic children differs to that previously reported in autism, this suggests that motion processing difficulties are not a general marker of atypical development (e.g., Braddick et al., 2003), but are instead condition-specific". We have also included references to previous studies comparing autistic and dyslexic populations in motion coherence tasks (Pellicano & Gibson, 2008; Tsermentseli et al., 2008) to provide additional context (p. 6).

We also note that autism is mentioned in relation to hypothesis 2 (see p. 6 and p. 17), as we consider two possibilities: one is that dyslexic children will show decreased sampling compared to typically developing children (in line with previous explanations), while the second is that dyslexic children will show a similar pattern of enhanced performance as in autistic children (Manning et al., 2015, 2017).

3. Assessing reading in controls

I appreciate that some of the control data has already been collected but I feel that failing to collect reading and spelling data in controls is a flaw in the design. It's critical to demonstrate that the groups differ on these key abilities in order to draw the inferences we need from this work. In addition, I think it will be a missed opportunity if we aren't able to relate the perceptual performance back to reading/spelling abilities in a correlational/regression analysis. Including these measures and registering an analysis would add significant value to the project.

RESPONSE: We agree that ideally we would have reading and spelling data in all of the typically developing children. However, we have chosen to re-use data from typically developing children for whom we did not collect this data in order to avoid unnecessarily testing children and the considerable time and resources this entails. Following the reviewer's comment, however, we have decided to collect reading and spelling data on new typically developing children. We have reordered our description of Participants and justified the re-use of previous data on p. 8.

One reason for collecting this data in typically developing children is to ensure that no children have undiagnosed literacy difficulties. Collecting reading and spelling tests on the 19 new typically developing children will allow us to ensure that these participants do not have spelling and reading difficulties defined as a composite score of 89 or below. For the remaining children, we will rely on the parent responses to the background questionnaire. We note that in a recently completed study with similar recruitment methods where we collected reading and spelling data (Toffoli et al., in prep), the reading and spelling data corresponded well with literacy difficulties reported by parents. We will conduct follow-up analyses to assess whether any group effects we find in the full sample are also found in the subsample of participants for whom we have reading and spelling data for – although we note that this analysis may be underpowered, which we will consider when interpreting these results (p. 15).

The second reason for collecting reading and spelling data is to run correlational analyses relating reading and spelling abilities to task performance. Yet our pre-registered hypotheses focus on group differences, rather than individual differences. We agree that collecting this information is useful for conducting further analyses – and collecting reading and spelling data on the newly recruited typically developing children will increase the subset for whom this analysis is possible. However, we consider these analyses to be exploratory, and think that the current study which pre-registers 6 hypotheses is already complex enough without adding additional hypotheses regarding individual differences.

4. Page 3, lines 5/6: Hyphenation – e.g., motion-coherence thresholds - I've found that liberal use of hyphens for compound adjectives can be really helpful as a reader. Please consider adding a few throughout the manuscript.

RESPONSE: We have added in hyphens throughout the manuscript (e.g., for “motion-averaging”, “orientation-averaging”, “motion-coherence” and “orientation-coherence”).

5. Page 5, lines 43/44 “...we will be able to not only understand...” I think ‘better understand’ would be more accurate. This sounds like a solid study, but I'm not sure it'll explain everything!

RESPONSE: We have amended this (p. 5).

6. Page 5, autism: It'd be good to include more about why it's interesting to compare dyslexic and autistic observers – what's the theory here?

RESPONSE: The dorsal stream vulnerability theory proposes that motion processing is affected across a range of developmental conditions (see p. 4), including autism and dyslexia. On pp. 5-6 we now a) clarify that we will make comparisons between performance

relative to typically developing children, b) give further justification of why it is interesting to compare these two conditions, and c) give additional references to provide context.

7. Page 7, Power analysis: Useful to confirm which analysis this was based on?

RESPONSE: We only have an effect size estimate from studies comparing dyslexic and typically developing participants in the motion-coherence task, not the averaging tasks, so we have based the power analysis on this (see p. 7). However, all of our hypotheses are independent samples t-tests, so the power analysis applies to all of our hypotheses. We have now clarified this on p. 7 (“in a two-tailed independent samples t-test”) and in the Study Design Template (pp. 16-21).

8. Page 8, 8 to 14 years of age: This is a broad age-range. Please include a justification for this - there’s a lot of general cognitive development going on across these ages. Will this be factored into the analysis? Was this included in the power analysis?

RESPONSE: Having a relatively wide age range allows us to reach the sample size required for a well-powered study – narrowing the age range would make this considerably more difficult. We have now noted this on p. 7. The power analysis did not include age (it was based on an independent samples t-test between groups, as we now state explicitly).

Following our studies of typical development, we do expect some age-related differences within this age range (see Manning et al., 2014, Figure 3). However, we do not yet have any hypotheses regarding different rates of development in dyslexia and typical development. This study is the first to apply the equivalent noise paradigm to a dyslexia population, and will help determine the presence and/or absence of group differences. Following this, future studies will be able to investigate whether dyslexic and typically developing children follow different trajectories in this task. We can look at possible age-related effects in our sample in exploratory analyses, but we may be underpowered to determine how age relationships differ between the groups, given that there is considerable variability even within an age group (see again Manning et al., 2014, Figure 3).

9. Page 8, composite score of 89 or less: This seems like quite a high cut off. I appreciate you’ve referenced Snowling’s work for this but it would be useful to include the justification here. Typically I’d expect to be below one standard deviation which I assume would be 85 for a test like this.

RESPONSE: The reviewer is correct that 100 is the mean and 85 is one standard deviation below the mean of these standardised tests for the whole population. However, in practice for opt-in studies such as this, the mean of the typically developing group tends to be higher than 100. For example, Snowling et al. (2019a, 2019b) reported a mean of 106.88 and SD = 11.68 in their typically developing group, and in a recently completed study (Toffoli et al., in prep), our group of typically developing children had a mean composite score of 107.31 and SD = 12.88. The cut-off of 89 was chosen by Snowling et al. (2019) as 1.5 standard deviations below the typically developing mean. We have now explained this on p. 8: “note the cut-off score of 89 was chosen to correspond to 1.5 standard deviations below the mean of typically developing children in a similar study (Snowling et al., 2019a, 2019b)”.

10. Page 8, speeded phonemic decoding + spelling

> Extremely useful to include information/discussion of subtypes in the introduction. Some references to consider (Castles & Coltheart, 1993; Jones et al., 2011; Kohnen et al., 2012; McArthur et al., 2013). You might also want to include the sight word reading subscale of the TOWRE to help pinpoint sub skills (i.e., Coltheart et al., 2001; Ziegler et al., 2008). And, just to be sure that you know about it, there’s a freely available non-speeded reading test of these subskills (Castles et al., 2009).

RESPONSE: As outlined above, we now mention subtypes in the introduction (p. 4). Children will complete both the sight word reading and phonological decoding efficiency subtests to characterise the sample, although only the latter will be used to form the composite score for inclusion/exclusion purposes. We now clarify the use of both subtests on p. 12.

11. Page 7, measure reading and spelling in controls – I think it’s super important to know how your controls perform on these tasks. We really need to be able to establish the group differences but it would be useful to be able to relate these skills back to your perceptual tasks (i.e., with correlations or regressions).

RESPONSE: As explained above, we now plan to collect reading and spelling data from newly collected typically developing children (see p. 8 and p. 12).

12. Page 8, exclusions with replacement on perceptual tasks? Just wanted to check whether there’s the intention to recruit more individuals with dyslexia (and controls) if they’re excluded from the perceptual tasks.

RESPONSE: Yes – if we need to exclude children from the dataset for any of the reasons provided (including poor performance in the motion tasks, see p. 9), we will replace them. We have now clarified this on p. 7: “The final dataset (following exclusions) will include 48 dyslexic children and 48 typically developing children”.

13. Page 10, keyboard responses from children for reaction times? There would be benefit in having children press buttons/keys for their responses. It’ll likely save time but also allow for reaction time to be included in the data set. This may be informative for other analyses of the data, but could even be factored in as a covariate. Given that the youngest children will be 8, this should be easily achieved.

RESPONSE: We see the advantages of collecting response time data in allowing additional analyses of the data. In our previous study (where we had children as young as 6 years old), we had an experimenter initiating trials (to ensure the child was attending) and collecting responses for children. Given that we will be re-using the data from some of the typically developing children, we have decided to retain this aspect of the procedure. In addition, this will allow us to make comparisons with the autism dataset. Otherwise, any differences in the pattern of results could be attributed to different motor/response demands.

14. Page 11, new reading/IQ data - reading, as well as general capacity, can vary a lot across time. Given that the TOWRE-2 has 4 (or more?) parallel forms, I’d strongly encourage you to collect up-to-date data on this. To best examine the relationship between reading and perception, the closer together in time the measurements are taken the better.

RESPONSE: Our research aims and hypotheses are not designed to examine relationships between reading and perception, but to conduct between-groups comparisons. We remain concerned about over-testing children and practice effects for the WASI and the WIAT-spelling (we note that dyslexic children are often given these tests regularly in education settings too), but we agree that the TOWRE-2 can be re-assessed due to the fact that there are 4 parallel forms so we will do this (see p. 12: “New TOWRE-2 scores will be collected using an alternate form than used in the previous study”).

15. Page 12, threshold estimates - apologies if I missed this but I wasn’t sure how the thresholds were calculated. This is typical based on the final X reversals of a staircase. It would be good to clearly define this (sticking with an even number of reversals to avoid bias)

RESPONSE: QUEST returns the most probable Bayesian estimate of the threshold at the end of the procedure using the mean or mode of the posterior probability density function (Watson & Pelli, 1983). We use the mean of the posterior probability density function, following previous studies using the efficient version of the equivalent noise model (e.g., Tibber et al., 2014, 2015; Manning et al., 2015). We now specify this in the methods section (p. 13). We have now also added details of the priors of the QUEST function on p. 11.

16. Page 14, replacing extreme values - One of the characteristics of perceptual performance in dyslexia is extreme values (Roach et al., 2004). I appreciate the up-front specification of these adjustments for pre-registration but I am concerned that we might be ‘throwing the baby out with the bath water’. Please consider non-parametric alternatives for the analyses – perhaps just as a contingency plan if that data are not normally distributed.

RESPONSE: This is a good point. We had originally proposed to keep the screening and transformation the same as in our autism studies, but given that we are not planning statistical analyses to directly compare autistic and dyslexic children, we have now changed this. We have removed the section on Data screening and transformation and instead state under Statistical Analysis that we will use non-parametric equivalents (p. 15).

17. Page 14, statistical analyses – what about autism? Although none of the hypotheses speak to autism, comparisons are mentioned in the introduction. Given that this is a registered report, I think it’s be useful to specify how these comparisons will be treated.

RESPONSE: Thank you for asking us to clarify this. Given that we do not yet know the pattern of performance that dyslexic children will show compared to typically developing children, it is difficult to specify follow-up statistical analyses to directly compare the developmental conditions, and the age ranges of the participants in the autism and dyslexia groups will be slightly different. Instead, we envisage making comparisons between the two studies (e.g., autistic children show enhanced motion integration compared to typically developing children, whereas dyslexic children do not) in the Discussion section, for example by comparing effect sizes or mean differences. We have now clarified this on pp. 5-6 (“to compare patterns of performance in dyslexic and autistic children relative to typically developing children”). As an example of this, under hypothesis 2 (p.6 and p. 17), you will see that we consider two possibilities: first, that dyslexic children will show decreased sampling compared to typically developing children (in line with previous explanations), and second, that dyslexic children will show a similar pattern of performance (i.e., increased sampling compared to typically developing children) as in autistic children.

It also feels remiss to not make some predictions (or plan to explore) about the relationships between reading and perception. Ideally this would be conducted across the entire dataset (i.e., dyslexic and typical readers) but this hinges on collecting their reading data.

RESPONSE: The aim of this study is to assess group differences in perception, for which the registered report already includes six registered hypotheses. We do not have any strong hypotheses for relationships between individual differences in reading and perception beyond simply extending the between-groups hypotheses into a correlation. We will not have reading and spelling scores for all of the typically developing children, but we agree that this could be an interesting exploratory analysis (either for this study or follow-up work). Following the guidelines for Registered Reports, we avoided describing exploratory analyses in this Stage 1 Report where possible, as it is deemed to blur the distinction between exploratory and confirmatory analyses.

Additional references not included in the manuscript

- Kover, S. T., & Atwood, A. K. (2013). Establishing equivalence: Methodological progress in group-matching design and analysis. *American Journal on Intellectual and Developmental Disabilities, 118*(1), 3-15.
- Manning, C., Dakin, S. C., Tibber, M. S., & Pellicano, E. (2014). Averaging, not internal noise, limits the development of coherent motion processing. *Developmental Cognitive Neuroscience, 10*, 44-56.
- Protopapas, A. (2019). Evolving concepts of dyslexia and their implications for research and remediation. *Frontiers in Psychology, 10*, 2873.
- Sassenhagen, J., & Alday, P. M. (2016). A common misapplication of statistical inference: nuisance control with null-hypothesis significance tests. *Brain and language, 162*, 42-45.

Appendix B

Dr Catherine Manning
School of Psychology and Clinical Language Sciences
University of Reading

Tel +44 (0) 118 378 3454
Email c.a.manning@reading.ac.uk

9th February 2022

Professor Chris Chambers
Subject Editor, *Royal Society Open Science*

Dear Professor Chambers,

Please find enclosed a Stage 2 Registered Report titled *Integration of visual motion and orientation signals in dyslexic children: An equivalent noise approach* (RSOS-200414) that we would like to be considered for publication at *Royal Society Open Science*.

The study data and digital materials/code are provided at <https://osf.io/76w59/> (see p. 22 in the manuscript). The approved Stage 1 protocol can be found at <https://osf.io/76w59/registrations> (see p. 7 in the manuscript). No other data other than that reported at Stage 1 was collected prior to the date of in-principle acceptance (see p. 9, line 216-223).

The completed experiment has been executed and analysed in the manner originally approved, with any unforeseen changes clearly noted. As we discussed with the Editor, we followed COVID-19 control measures mandated by the institution which were put in place after in-principle acceptance was granted (see p. 14 for description). No other changes were made to the approved methods and analyses.

We have highlighted in red font any edits to the Introduction, Methods and Abstract section. An additional author has been added (Victoria Hulks) as this was the Research Assistant who collected the majority of the data, and she meets the criteria for authorship. Note changes in the Introduction include tense changes only.

We look forward to hearing from you again.

Yours sincerely,

Catherine Manning (on behalf of the authors)

Appendix C

Dr Catherine Manning
School of Psychology and Clinical Language Sciences
University of Reading

Tel +44 (0) 118 378 3454
Email c.a.manning@reading.ac.uk

17th March 2022

Professor Chris Chambers
Subject Editor, *Royal Society Open Science*

Dear Professor Chambers,

Please find enclosed a revised Stage 2 Registered Report titled *Integration of visual motion and orientation signals in dyslexic children: An equivalent noise approach* (RSOS-200414.R3). We were pleased to receive positive comments from you and the Reviewer. We have responded to the comments point-by-point below (see changes highlighted in yellow). We hope that you will find this version suitable for publication at *Royal Society Open Science*.

We look forward to hearing from you again.

Yours sincerely,

Catherine Manning (on behalf of the authors)

ASSOCIATE EDITOR COMMENTS

One of the original reviewers from Stage 1 kindly returned to evaluate the Stage 2 manuscript, and I have decided that we can continue with an interim decision based on this reviewer's assessment and my own reading of the manuscript. As will see, the reviewer is broadly very positive about your submission and notes some minor areas that would benefit from elaboration or at least further consideration in a response. I concur and agree that this is a well conducted and reported RR, under challenging circumstances.

I will assess the revised manuscript and response at desk, and provided you are able to respond to all points thoroughly then I anticipate that full acceptance should be forthcoming without requiring further in-depth review.

RESPONSE: Thank you to you and the reviewer for carefully reading this submission and for your positive comments.

REVIEWER 2 COMMENTS

Dear Cathy and colleagues,

Firstly, well done for completing this project during COVID times. It's impressive in general in terms of your timeframe, but even more so under the circumstances. As

mentioned below, I was surprised by the small magnitude of the effects, but that's one of the benefits of preregistration. The updates are nicely presented and clear. I have a few comments for consideration below – always happy to clarify/discuss over email if that's helpful.

Best wishes,
Nic Badcock

Required comments:

1. Whether the data are able to test the authors' proposed hypotheses by passing the approved outcome-neutral criteria (such as absence of floor and ceiling effects or success of positive controls)

> I confirm that the data are suitable to test the hypotheses and there is sufficient variability in the data

2. Whether the Introduction, rationale and stated hypotheses are the same as the approved Stage 1 submission

> This is consistent – modifications for past tense have been made as appropriate

3. Whether the authors adhered precisely to the registered experimental procedures

> I confirm this to be the case

4. Where applicable, whether any unregistered exploratory statistical analyses are justified, methodologically sound, and informative

> The additional analyses are a useful inclusion, clarifying some pertinent queries related to the sampling with respect to previous work

5. Whether the authors' conclusions are justified given the data

> The conclusions fit nicely. As I'll note below, it would be useful to temper the absolute nature of significance with the subtleties of effect sizes and inconclusive/weak Bayesian evidence.

RESPONSE: Thank you for your positive comments on our Stage 2 manuscript. Regarding point 5, we used Table 2 to ensure our conclusions/interpretations matched those that we had pre-registered (i.e., interpreting a significant p-value as support for our hypotheses), but we have now added more discussion about the small effect sizes and weak levels of Bayesian evidence (see response below).

General comments:

6. Magnitude of effect

One of the great benefits to the preregistration is that we get to see that magnitude of effects in a transparent investigation. I was surprised by the small effect sizes reported, especially within the context of the Cohen's d value used for the power analysis. This surprise is in no way a criticism – I imagine it reflective of the true effects in a heterogenous sample, as you note in the Discussion. I would like to see the magnitude of the effects highlighted in the Abstract and Discussion to a higher degree. I feel/fear that the black and white nature of NHST is too extreme in this case, particularly in the Abstract, and including a comment on the nature of the effect size of the significant effects, and Bayesian evidence if it fits within the word count, would do better service to the data.

To be sure, this magnitude is reflected in the Discussion (e.g., line 532: "...to note that the group differences were subtle..." and 574: "...the relatively inconclusive nature of our results...") but it may be worth having an explicit sentence or two in the opening paragraph dedicated to confirming the small magnitude of effects in relation to the basis for the power analysis. It could be flagged as a limitation if you see fit – again, not so much a criticism of the study, but the literature upon which it was based.

RESPONSE: When interpreting the results of the p-values and Bayes factors in the abstract and discussion, we referred to our pre-registered Table 2 which stated that we

would interpret a significant p-value in NHST as support for our hypothesis – and that in the event of non-significant p-values, that our conclusion would be based on the results of the Bayes factors. We therefore are wary of changing our interpretation as this would deviate from the pre-registration. However, in line with the Reviewer’s comments (and ensuring we still comply with the word limit), we have now added a statement into the abstract: “*While group differences were subtle*, dyslexic children had significantly higher internal noise estimates for motion discrimination, and higher orientation coherence thresholds, compared to typical children”. The first paragraph of the Discussion already accurately represents the pattern of data (in line with the pre-registered interpretations shown in Table 2), but we have now added a qualifying statement when we report the significant results: “although these group differences were subtle” (p. 28, line 509), and added clarification in response to point 12 below. Regarding the relationship to the power analysis, we note that the non-parametric and Bayesian analyses do not lend themselves to direct comparison with the effect sizes (Cohen’s *d*) used in the power analysis.

7. COVID considerations

This is a very light comment, but I wondered whether there’s any sense in the literature regarding sampling biases with respect to who’s willing to participant in research during a pandemic. My impression is that the UK, certainly more so than Australia, has been business as usual, but it seems prudent to suggest a little check of the literature with an eye for a potential bias of whether individuals with more extreme difficulties (in reading, in general) may be avoiding testing during these times. As mentioned, a very light comment, I’m not aware of anything, but it could be worth a quick look. I note that the reading scores (Table 1) are as you might expect, and it’s hard to imagine that poor readers with particularly problematic magno deficits would somehow avoid testing – so perhaps it’s just worth a reminder in the limitations that the testing was conducted during COVID times which may have influenced the population willing to participate. Only if this seems at all relevant in your opinion.

RESPONSE: We also know of no reason why those with particularly extreme reading difficulties would be less likely to participate due to the pandemic – as this population isn’t defined as a clinically vulnerable group. We also haven’t found any relevant literature on this point. However, we now reflect on possible sampling biases as a result of COVID in the limitations paragraph in the Discussion section (p. 33, line 630-631): “A third limitation is that the new data were collected during the COVID-19 pandemic, and it is possible that this may have influenced the population willing to participate”.

Specific comments:

8. Page 6, line 147: “2. We investigated whether dyslexic children show atypical sampling in the motion- ...” > Please consider rephrasing this to be in line with the other hypotheses (i.e., We hypothesised that ...) – as included in Table 2.

RESPONSE: The pre-registered hypothesis was worded in this way because we were not clear of the direction to expect (there were reasons to expect either decreased or increased sampling) – unlike in hypotheses 1, 3, 4 or 5. In Table 2 the hypothesis is worded slightly differently, as this is the 2-tailed experimental hypothesis being tested by the statistical tests. We have consulted the Editor and been advised not to change the hypothesis, as this was the hypothesis that was accepted in principle.

9. Page 7, line 167: same comment as previous regarding item number 6. Apologies if I’ve missed something here. I guess either way, some clarification in the wording would help slow people like me 💎💎 (I also appreciate that these are comments related to the previously accepted portion of the manuscript – hopefully it’s okay to comment on this)

RESPONSE: As mentioned in response to point 8 above, this was because it was unclear whether we should predict increased internal noise in dyslexia, or no differences. We have retained the wording in line with the in-principle acceptance.

10. Page 22/23, Line 397 – 399: Table 1 header row > Consider adding an extra row for the column headers with M (SD) Error Range for the Typically Developing and Dyslexic columns – so twice. It’s a bit repetitive but I feel it’ll make the table more accessible. The note under the table feels less intuitive than additional column headers to me. Additionally, consider confirming in the table title that the error range reflects a portion – my first impression is that error range is often a count.

RESPONSE: We have now added an extra row to replace the note under the table for both Table 1 (p. 8) and Table 3 (pp. 22-23). We have also changed the title of Table 3 to reflect the fact that the errors are expressed as a proportion (“*Mean, standard deviation and range of proportion of errors made in catch trials*”). We have also clarified the text on p. 23 line 400 (replacing “*number of errors*” with “*proportion of errors*”).

11. Page 24, Figure 3 > Great to see all the data points on these plots! Please consider capitalising the group names on the x axes, just for presentation purposes, feels ever so slightly under baked with lower case (though I always prefer to code in lower case, so I appreciate where it comes from).

RESPONSE: We have now capitalised the group names on the x-axes for Figures 3 and 4.

12. Page 28, lines 512/513: “This analysis revealed inconclusive evidence for or against the remaining hypothesised group differences.” > my reading is that ‘the remaining’ might be better replaced by ‘all’ in this instance. Please consider whether this might fit.

RESPONSE: We have considered this suggestion carefully. However, the way we have summarised the results relates to the pre-registered analysis plan (see Table 2) – where we first planned to assess whether there were any significant results ($p < .05$), and in the case of non-significant results, our interpretation would be based on the results of Bayesian analyses. We have now clarified this in the relevant part of the discussion: “to quantify the relative evidence for the null and alternative hypotheses *to make inferences when non-significant differences were obtained*. This analysis revealed inconclusive evidence for or against the hypothesised group differences *for the measures in which non-significant results were obtained*” (p. 28 line 511-514).” The following sentence then goes on to explain that the Bayesian evidence was also not strong/conclusive in the case of the significant differences (for motion internal noise estimates and orientation coherence threshold estimates).

13. Page 32, line 621: > Could you comment on how your age distribution compares to other research where bigger effects were found?

RESPONSE: Previous studies have used a range of different age distributions (see Benassi et al., 2010, Table 1), but we now comment on this in the Discussion: “The wide age range could potentially have obscured group differences, although Benassi et al.’s (2010) meta-analysis (which we based our power analysis on) included studies which used similarly wide age ranges and found large group differences in motion coherence thresholds (e.g., Sperling et al., 2006; Slaghuis & Ryan, 2006).” (p. 32 line 626- p. 33 line 631).